# ANYDA: ANYTIME DOMAIN ADAPTATION

**Omprakash Charkraborty**[1]    **Aadarsh Sahoo**[2]    **Rameswar Panda**[2]    **Abir Das**[1]
[1]IIT Kharagpur, [2]MIT-IBM Watson AI Lab
{opckgp@,abir@cse.}iitkgp.ac.in, {aadarsh,rpanda}@ibm.com

## ABSTRACT

Unsupervised domain adaptation is an open and challenging problem in computer vision. While existing research shows encouraging results in addressing cross-domain distribution shift on common benchmarks, they are often constrained to testing under a specific target setting, limiting their impact for many real-world applications. In this paper, we introduce a simple yet effective framework for anytime domain adaptation that is executable with dynamic resource constraints to achieve accuracy-efficiency trade-offs under domain-shifts. We achieve this by training a single shared network using both labeled source and unlabeled data, with switchable depth, width and input resolutions on the fly to enable testing under a wide range of computation budgets. Starting with a teacher network trained from a label-rich source domain, we utilize bootstrapped recursive knowledge distillation as a nexus between source and target domains to jointly train the student network with switchable subnetworks. Experiments on multiple datasets well demonstrate the superiority of our approach over state-of-the-art methods. [1]

## 1 INTRODUCTION

Unsupervised Domain Adaptation (UDA) which aims to adapt models trained on a labeled *source* domain to an unlabeled *target* domain has attracted intense attention in recent years. However, recent successful UDA approaches (Carlucci et al., 2019; Ganin et al., 2016; Li et al., 2020a; Prabhu et al., 2021; Sun et al., 2019; Tan et al., 2020; Tzeng et al., 2015; 2017) often rely on complicated network architectures and are limited to testing under a specific target setting, which may not be particularly suitable for applications across a wide range of platforms that present different resource constraints (see Figure 1a). While adapting the trained model independently for all testing scenarios in the target domain with drastically different resource requirements looks like a possible option at the first glance, it is not efficient and economical, because of time-consuming training and benchmarking for each of these adaptation settings. Preferably, we want to be able to adjust the model, without the need of re-training or re-adaptation in the target domain, to run in high accuracy mode when resources are sufficient and switch to low accuracy mode when resources are limited.

Motivated by this, in this paper, we investigate the problem of *anytime domain adaptation* where we have labeled training data from a source domain but no labeled data in the target domain and in addition testing at a resource setting with wide range of variation (*e.g.*, see Figure 1b). Specifically, we aim to train a single network using both labeled source and unlabeled target data that can directly run at arbitrary resource budget while being invariant to distribution shifts across both domains. This is an extremely relevant problem to address as it will provide a distinct opportunity for a more practical and efficient domain adaptation to favor different scenarios with different resource budgets.

Recently, anytime prediction (Cai et al., 2019; Huang et al., 2018; Jie et al., 2019) that train a network to carry out inference under varying budget constraints have witnessed great success in many vision tasks. However, all these methods assume that the models are trained and tested using data coming from some fixed distribution and lead to substantially poor generalization when the two data distributions are different. The twin goals of aligning two domains and operating at different constrained computation budgets bring in additional challenges for anytime domain adaptation.

To this end, we propose a simple yet effective method for anytime domain adaptation, called **AnyDA**, by considering domain alignment in addition to varying both network (width and depth) and input

---

[1]Project page: https://cvir.github.io/projects/anyda

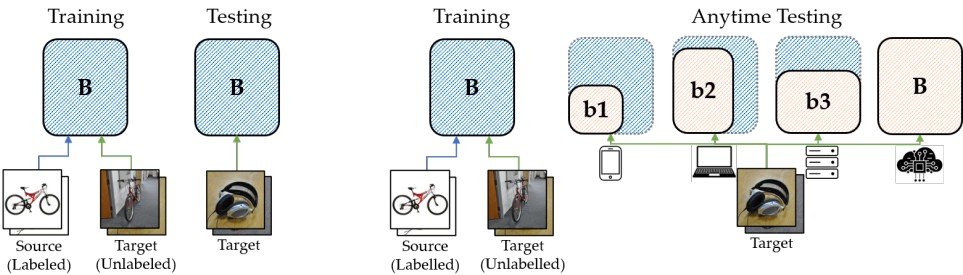

| (a) Unsupervised Domain Adaptation | (b) Anytime Domain Adaptation |

Figure 1: Instead of conventional adaptation under a specific computation budget, anytime domain adaptation focuses on training a model using both labeled source and unlabeled target data that can directly run at arbitrary resource budget in the target domain while being invariant to distribution shifts across both domains.

(resolution) scales to enable testing under a wide range of computation budgets. Such variation over width, depth and resolution enables tighter as well as finer coupling of efficiency-computation trade-off than prior works that only focus on one or two out of the three dimensions (Li et al., 2021a; Yang et al., 2020; Yu et al., 2019b) for in-domain data. In particular, we adopt a switchable network where individual subnetworks executable at variable computation budget share parameters with the full-budget network, known as the *supernet*. However, inability to leverage on the higher capacity of complex networks may, in effect, cause such a model to severely underperform, leading to sub-optimal performance across different resource budgets. To alleviate this, we propose to distill (Ba & Caruana, 2014; Hinton et al., 2015) richer alignment information from higher capacity models to networks with limited computation. In particular, our proposed **AnyDA**, adopts two switchable networks as teacher and student, that interact and learn from each other. The student subnetworks are trained recursively to fit the output logits of an ensemble of larger subnetworks of the teacher. Such recursive distillation within a single network with only adaptive width is shown to improve generality and reduce performance gaps between high and low capacity networks (Li et al., 2021a).

Starting with the labeled source data, we build the teacher from past iterations of the student network as an exponential moving average (ema) of the student. The bootstrapped teacher provides the targets to train the student for an enhanced representation. Once the target data is available, the *bootstrapped recursive distillation* not only brings the target features close to the source but also transfers the learned knowledge to a smaller network for efficient inference. Moreover, we harness the categorical information by leveraging self-supervision through a pseudo-label loss on the student supernet to ensure a discriminative latent space for the unlabelled target images. Interestingly, without using any component for explicit domain alignment (*e.g.*, a domain discriminator), we show that our approach trades the performance gracefully with decreasing budgets of the subnetworks. Our extensive experiments on 4 benchmark datasets show very minimal drop in performance across a wide range of computation budget (*e.g.*, a maximum drop of only $1.1\%$ is observed when the range of computation budget gets $8\times$ small during testing in the Office-31 dataset (Saenko et al., 2010)).

Our work forges a connection between two literatures that have evolved mostly independently: *anytime prediction* and *domain adaptation*. This connection allows us to leverage the progress made in unsupervised representation learning to address the very practical problem of anytime domain adaptation. To summarize, our key contributions include:

- We introduce a novel approach for anytime domain adaptation, that is executable with dynamic resource constraints to achieve accuracy-efficiency trade-offs under domain-shifts. We achieve this by training two networks as teacher and student with switchable depth, width and input resolutions to enable testing under a wide range of computation budgets.

- We propose a bootstrapped recursive distillation approach to train the student subnets with the knowledge from the teacher network that not only brings the target features close to the source but also transfers the learned knowledge to a smaller network for efficient inference.

- We perform extensive experiments on 4 benchmark datasets and demonstrate that **AnyDA** achieves superior performance over the state-of-the-art methods, more significantly at lower computation budgets. We also include comprehensive ablation studies to depict the importance of each module of our proposed framework.

## 2 RELATED WORKS

**Efficient Domain Adaptation.** UDA has been dominated by methods minimizing some measure of domain discrepancy (Shen et al., 2018; Sun & Saenko, 2016; Tzeng et al., 2015) or maximizing domain confusion (Ganin et al., 2016; Long et al., 2018; Pei et al., 2018; Tzeng et al., 2017) to generate domain-invariant features. Leveraging image translation (Hoffman et al., 2018; Murez et al., 2018) or style transfer (Dundar et al., 2020; Zhang et al., 2018) is also another popular trend. Self-supervised approaches like solving pretext tasks (Carlucci et al., 2019; Mei et al., 2020; Sahoo et al., 2023; Sun et al., 2019) and contrastive learning (Li et al., 2020a; Prabhu et al., 2021; Sahoo et al., 2021; Tan et al., 2020) have also recently enjoyed huge success in aligning domains. In spite of the growing development of traditional UDA (Kouw & Loog, 2019; Zhao et al., 2020), the challenging problem of *efficient domain adaptation* remains largely underexplored. Authors in (Jiang et al., 2020; Li et al., 2021b) have proposed a multi-scale early-exit architecture (Huang et al., 2018) with DANN (Ganin et al., 2016) as the domain adaptation method. We, on the other hand, employ recursive knowledge distillation from an ensemble of teacher subnetworks that produces consistency supervision for the student subnetworks. The proposed approach eliminates the need for resource hungry spatial augmentation and further reduces the training burden by eliminating the domain discriminator which is unwieldy to train. The contemporary work SlimDA (Meng et al., 2022), attaches a weight-sharing slimmable network to a domain symmetric adaptation network (SymNet) (Zhang et al., 2019b). The stochastic distillation uses an empirical confidence score for the subnets requiring the bi-classifier of SymNet to be replaced by a tri-classifier. Ours, on the other hand, is model agnostic showing strong transferability across backbones. In addition, unlike early-exit architectures with varying depth only or SlimDA with varying width only, we train a network with switchable depth, width and input resolution. To the best of our knowledge, this work is the first attempt to address efficient domain adaptation under anytime prediction framework through switchable depth, width and input resolution with domain specific batch normalization.

**Anytime Neural Networks.** Anytime neural networks (Hu et al., 2019; Jie et al., 2019; Larsson et al., 2017) are becoming increasingly attractive due to its computational efficiency. While MS-DNet (Huang et al., 2018) makes early exits to meet varying resource demands, MutualNet (Yang et al., 2020) trains a single network to achieve accuracy-efficiency tradeoffs at runtime. Skipping unimportant channels (Chen et al., 2019) and layers (Wang et al., 2018; Wu et al., 2018), dynamic routing of multiple inference paths (Liu & Deng, 2018; Li et al., 2020b; McGill & Perona, 2017) and input dependent adaptation (Chen et al., 2020) are investigated for efficient inference on many applications. Slimmable networks (Yu et al., 2019b) and it's variants (Yu & Huang, 2019; Li et al., 2021a) train a model to support different width multipliers. Adjusting width, depth and kernel sizes simultaneously to achieve better accuracy-efficiency trade-off is also proposed in (Cai et al., 2019; Han et al., 2020). Typically, anytime prediction works under the assumption of no distribution shift between train and test data. While our approach is inspired by these, in this paper, we propose a framework that provides budget adaptive anytime predictions under change of domains.

**Knowledge Distillation.** Distilling knowledge by mimicking output (Hinton et al., 2015; Buciluǎ et al., 2006) and intermediate features (Romero et al., 2014) from an ensemble of teachers or self-distilling (Furlanello et al., 2018; Ji et al., 2021; Lee et al., 2020; Zhang et al., 2019a) knowledge from previous iterations have obtained compact models without sacrificing the performance. Recent works have explored self-supervised learning with knowledge distillation (Caron et al., 2021; Fang et al., 2021; Tian et al., 2020; Xu et al., 2020) enabling model compression and performance gains without explicit supervision. However, these works rely on a teacher trained on data from the same domain while our teacher is bootstrapped from the student that is trained with both source and target data stabilizing the feature alignment process between the two domains.

## 3 METHODOLOGY

Anytime domain adaptation aims to achieve better generalization and low inference latency across different platforms in unlabeled target domain by transferring knowledge from a labelled source domain. Formally, we have a set of $N_s$ labelled source images $\mathcal{D}_s = \{(x_s^i, y_s^i)\}_{i=1}^{N_s}$ and a set of $N_t$ unlabelled target images $\mathcal{D}_t = \{x_t^i\}_{i=1}^{N_t}$, with a common label space $\mathcal{L}$. The source and target samples have different data distributions. We also have a set of $n$ computation budgets $\mathcal{B} = \{b_1, b_2, ..., b_n\}$ for inference with $b_i < b_j$ when $i < j$. Our goal is to learn a single domain adaptive model executable at a wide range of computation budget on the target domain with minimal performance drop.

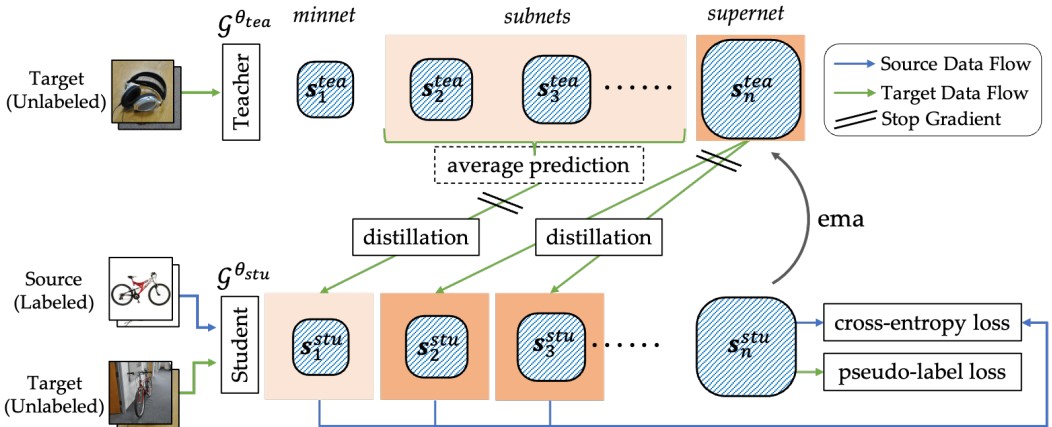

Figure 2: **Illustration of our approach.** We use a teacher-student framework consisting of 2 networks. The student network $\mathcal{G}^{\theta_{stu}}$ takes both source and target images as input, while the teacher network $\mathcal{G}^{\theta_{tea}}$ takes only target images. Both networks are configured into $n$ subnets with increasing budgets. We recursively distill knowledge from the teacher network to train the student subnets for anytime domain adaptation and bootstrap on previous representations with ema update for the teacher network. During inference, only student network is used by selecting the appropriate subnet adhering to the budget constraints. See Section 3 for more details.

## 3.1 PRELIMINARIES

**Budget Configurations.** To make a single network executable at different budgets, we consider degrees of freedom both at the level of input (*resolution*) and network (*width*, *depth*), denoted by $r$, $w$, and $d$ respectively. $r$ takes positive integral values, while $w$ and $d$ are numbers in the interval $(0, 1]$. Let $\mathcal{R} = \{r_1, r_2, ..., r_a\}$, $\mathcal{W} = \{w_1, w_2, ..., w_b\}$, and $\mathcal{D} = \{d_1, d_2, ..., d_c\}$ denote the sets of possible values of $r$, $w$, and $b$. Following (Yang et al., 2020; Yu et al., 2019b), given a configuration tuple $<r_x, w_y, d_z> \in \mathcal{R} \times \mathcal{W} \times \mathcal{D}$ we obtain a subnetwork (or *subnet*) by executing only the first $w_y \times 100\%$ filters and $d_z \times 100\%$ blocks in each layer and take input images of $r_x \times r_x$ resolution consequently utilizing a specific budget $b_i \in \mathcal{B}$. Let the set of all subnets for a network $\mathcal{G}$ be denoted as $\mathcal{G}^* = \{s_1, s_2, ..., s_n\}$, where $s_i$ is a *subnet* with budget $b_i$. The network with the highest computation budget executable at the highest width, depth and input resolution is denoted as the *supernet*, while the one with the lowests is called the *minnet*. Rest of the networks are simply addressed as *subnet*. We compute the budget in terms of Multiply-Accumulate operations (MAC), following (Li et al., 2021b), and provide the detailed budget calculation for a given tuple in Table 4 of the Appendix F.

**Domain-Specific Switchable Batch Normalization.** Independently reducing internal covariate shift through batch normalization (Ioffe & Szegedy, 2015) for each of the subnets is crucial for learning conditional information specific to the network configurations (Yu et al., 2019b). Additionally, batch normalization parameters have been used to encode domain-specific information for unsupervised domain adaptation (Chang et al., 2019). For these reasons, in `AnyDA`, we learn batch normalization statistics individually for each domain as well as for each of the subnets.

## 3.2 APPROACH OVERVIEW

Figure 2 provides an overview of our `AnyDA` approach. We adopt a teacher-student framework consisting of two switchable networks with the same architecture, namely $\mathcal{G}^{\theta_{stu}}$ and $\mathcal{G}^{\theta_{tea}}$ parameterized respectively by $\theta_{stu}$ and $\theta_{tea}$, which are configured into $n$ subnets with increasing budgets following $\mathcal{B}$. The student network is fed with both source and target images, while the teacher network takes only target images. Given an input image, each of the student subnets $s_i^{stu}$ and teacher subnets $s_i^{tea}$ predicts probability distributions $p_i^{stu}$ and $p_i^{tea}$ over the label space $\mathcal{L}$ by utilizing a maximum budget of $b_i$ per prediction. The student supernet, subnets, and minnet are trained using supervised learning on the source data. For target data, we propose a *bootstrapped recursive distillation* approach to train the student subnets with knowledge obtained from the teacher network. Specifically, the student minnet is trained to match its output with the average prediction of the teacher subnets, while the student subnets are trained to match their outputs with the prediction of the teacher supernet. Teacher network parameters are updated as exponential moving average of that of the student network. The teacher network only gets target data as input and provides the cross-domain knowledge essential for domain alignment to the lower computation budget student

networks. Additionally, to ensure a discriminative latent space for the unlabelled target images, we harness the categorical information by leveraging self-supervision through a pseudo-label loss on the student supernet. We now describe each of our proposed components individually in detail.

## 3.3 ANYTIME DOMAIN ADAPTATION

**Bootstrapped Recursive Distillation.** Self-distillation (Caron et al., 2021) leverages multiple views of the same input to train a student network to match the output of a teacher network. To this end, we propose a recursive distillation learning paradigm where we emulate multiple views of an input through the outputs of the budgeted subnets. Each of the individual subnets encodes unique information specific to the configured *width*, *depth* and *resolution* providing us with diverse knowledge for a given input. For each of the subnets in $\mathcal{G}^{\theta_{stu}*}$ and $\mathcal{G}^{\theta_{tea}*}$ the predicted logits are normalized with softmax for any input $x$ to output a probability distribution over the label space $\mathcal{L}$ as follows:

$$p_j^{stu(i)} = \frac{\exp(s_j^{stu(i)}(x)/\tau_{stu})}{\sum_{l=1}^{|\mathcal{L}|} \exp(s_j^{stu(l)}(x)/\tau_{stu})} \quad p_j^{tea(i)} = \frac{\exp(s_j^{tea(i)}(x)/\tau_{tea})}{\sum_{l=1}^{|\mathcal{L}|} \exp(s_j^{tea(l)}(x)/\tau_{tea})} \quad \forall i \in \mathcal{L} \quad (1)$$

where, $s_j^{stu}$ and $s_j^{tea}$ are any subnets in $\mathcal{G}^{\theta_{stu}*}$ and $\mathcal{G}^{\theta_{tea}*}$ respectively. $\tau_{stu}, \tau_{tea} > 0$ are temperature parameters controlling the sharpness of the output distributions. We allow the student network to observe data from both source and target domain while the teacher network sees data from target only. This is crucial to alleviate the distribution shift by encouraging "target-to-source" correspondences for distillation from the teacher to the student. Ensemble of teacher networks is shown to generate more accurate and general soft-labels for distillation (Shen et al., 2019; Shen & Savvides, 2020). So, we consider the average prediction of the teacher subnets $s_2^{tea}$ to $s_{n-1}^{tea}$ as a representative of the diverse knowledge learned by them. Additionally, this reduces the effect of noisy subnets while distilling. The average prediction is obtained by averaging the logits and then softmaxing. Now, for a target image $x_t \in \mathcal{D}_t$, let the corresponding softmaxed logits from the student subnets be $\{p_{t1}^{stu}, p_{t2}^{stu}, ..., p_{tn}^{stu}\}$, the same from the teacher subnets be $\{p_{t1}^{tea}, p_{t2}^{tea}, ..., p_{tn}^{tea}\}$ and $p_{tavg}^{tea}$ be the average prediction of the subnets $s_2^{tea}$ to $s_{n-1}^{tea}$. We formulate the recursive distillation loss as,

$$\mathcal{L}_{rd}(x_t) = \mathrm{H}(p_{t1}^{stu}, p_{tavg}^{tea}) + \sum_{i=2}^{n-1} \mathrm{H}(p_{ti}^{stu}, p_{tn}^{tea}) \quad (2)$$

where, $\mathrm{H}(a, b) = -\sum_{l=1}^{|\mathcal{L}|} a^{(l)} \log(b^{(l)})$ is cross-entropy loss and is minimized with respect to student parameters $\theta_{stu}$ only. The first term in Eqn. 2 trains student minnet to mimic average prediction of teacher subnets which helps to gain noise robustness using the ensemble especially when distilling to very low capacity networks. The second term encourages student subnets to mimic teacher supernet on target data, enabling a recursive flow of knowledge. The teacher parameters $\theta_{tea}$ are updated as: $\theta_{tea} \leftarrow \lambda \theta_{tea} + (1 - \lambda)\theta_{stu}$, where $\lambda$ represents the momentum hyperparameter. The teacher network is maintained as exponential moving average of historical parameters over training iteration allowing to bootstrap on previous representations, as in (Grill et al., 2020).

**Target Pseudo-Labels.** While the bootstrapped recursive distillation loss helps in learning low budget networks with simultaneous domain alignment, relying only on the source domain for categorical information can be sub-optimal for the target domain. Therefore, we use a pseudo-label loss on the student network to harness categorical information from the target domain through self-supervision. For a target image $x_t \in \mathcal{D}_t$ the pseudo-label loss is formulated as:

$$\mathcal{L}_{pl}(x_t) = \mathbb{1}(\max(p_{tn}^{stu}) \geq \tau_{pl})\mathrm{H}(\hat{y}_t, p_{tn}^{stu}), \quad \tau_{pl} : \text{ pseudo-label threshold} \quad (3)$$

where, $\hat{y}_t = \arg\max(p_{tn}^{stu})$ is the pseudo-label obtained from the student supernet $s_n^{stu}$ for $x_t$.

**Optimization.** While training, we follow the *sandwich rule* (Yu & Huang, 2019) for better convergence behavior and overall performance. Specifically, in every iteration, we train the student minnet, supernet, and *two randomly selected* subnets. This is inspired from the fact that performances at all budget are bounded by that of the model at smallest budget (minnet) and largest budget (supernet). Thus, optimizing performance lower bound and upper bound can implicitly optimize all subnetworks of different capacities. Additionally, this is computationally more efficient compared to training all the subnetworks in each iteration. Formally, at a given iteration with mini-batches $\beta_s \subset \mathcal{D}_s$ and $\beta_t \subset \mathcal{D}_t$, we optimize the following loss function:

$$\mathcal{L}_{total}(\beta_s, \beta_t) = \mathbb{E}_{x_s \in \beta_s} \lambda_{cls}\mathcal{L}_{cls} + \mathbb{E}_{x_t \in \beta_t} (\lambda_{rd}\mathcal{L}_{rd} + \lambda_{pl}\mathcal{L}_{pl}) \quad (4)$$

where, $\lambda_{cls}$, $\lambda_{rd}$, and $\lambda_{pl}$ are loss coefficients. $\mathcal{L}_{cls}$ denotes the cross-entropy loss on labeled source images on all the subnets (ref. Figure 2). Also, note that because of the sandwich rule, the second term of $\mathcal{L}_{rd}$ (ref. Eqn. 2) now applies on the two selected subnets only. Following this, the teacher network parameters are updated using ema in every iteration.

**Warm-up using Source Data.** Absence of labels in the target domain may quickly make the networks learn a degenerate solution if we do not take care with proper initialization (Tzeng et al., 2017). Additionally, random initialization is prone to provide high-confident noisy pseudo-labels leading to catastrophic outcomes. Therefore, before training **AnyDA** with Eqn. 4, we warmup the student network with the $\mathcal{L}_{cls}$ loss on the source data. Once the warmup is completed, we copy the learned weights $\theta_s$ to $\theta_t$ to provide the same initialization to both the networks.

**Inference.** Once the networks are trained, we only use the student network for inference. Given any computational budget $b_i \in \mathcal{B}$, we first obtain all possible configuration tuples in $\mathcal{R} \times \mathcal{W} \times \mathcal{D}$ forming subnets of budget $b_i$. E.g. in our experimental setup, given a budget constraint of $0.85 \times 10^9$ MACs, we have the tuples $< 224, 0.9, 0.5 >$ and $< 160, 0.9, 1.0 >$ satisfying the requirement. We then test all of these subnets on a validation set, choose the best performing subnet and report its accuracy.

## 4 EXPERIMENTS

**Datasets.** We evaluate the performance of our proposed approach using 4 benchmark datasets, namely, Office-31 (Saenko et al., 2010), DomainNet (Peng et al., 2019), Office-Home (Venkateswara et al., 2017) and ImageCLEF-DA (Long et al., 2017). Office-31 contains a total of $4,652$ images belonging to 31 categories from 3 distinct domains: Amazon (**A**), Webcam (**W**) and Dslr (**D**). DomainNet is the largest available benchmark dataset containing images from 6 domains: Infograph (**inf**), Quickdraw (**qdr**), Real (**rel**), Sketch (**skt**), Clipart (**clp**), and Painting (**pnt**). 600K images are distributed among $345$ categories. Office-Home is a challenging dataset containing images from 4 different domains: Artistic (**Ar**), Clipart (**Cl**), Product (**Pr**), and Real-World images (**Rw**) belonging to $65$ categories. The ImageCLEF-DA dataset has around 1,800 images belonging to 12 categories in 3 domains: Caltech-256 (**C**), ImageNet ILSVRC 2012 (**I**) and Pascal VOC 2012 (**P**).

**Baselines and Implementation Details.** We compare our approach with the following baselines: ResNet (He et al., 2016) models with varying depth (ResNet-18 to ResNet-152), MobileNetV3 (Howard et al., 2019), MSDNet (Huang et al., 2018), REDA (Jiang et al., 2020) equipped with DANN (Ganin & Lempitsky, 2015) as the domain adaptation method, TCP (Yu et al., 2019a) and ResNet (18, 34, and 50) trained with knowledge distillation (Ba & Caruana, 2014) from ResNet-101 as the teacher. We also compare with DDA (Li et al., 2021b). For the baselines and DDA, we quote the numbers reported in the paper itself except for OfficeHome and ImageCLEF-DA for which we use the publicly available source code. We use ResNet-50 (He et al., 2016) architecture as the full network. For budget configuration, we use $\mathcal{R} = \{224, 192, 160, 128\}$, $\mathcal{W} = \{1.0, 0.9\}$, and $\mathcal{D} = \{1.0, 0.5\}$, providing us 16 subnets with computational budgets ranging from roughly $0.2 \times 10^9$ to $2.0 \times 10^9$ MACs. We further divide the range of budgets into 8 intervals of equal size. For each interval, the best performing subnet is used to report the accuracy. Warmup using source data was done on top of networks initialized with Imagenet pretrained weights. In addition, following the general practice we use a validation set to obtain best hyperparameters. Additional implementation details including hyperparameter values are provided in the Appendix F.

### 4.1 RESULTS AND ANALYSIS

**Office-31.** Figure 3a shows the results on the Office-31 dataset. **AnyDA** achieves the best average accuracy of $85.1\%$ at a budget of $0.7 \times 10^9$ MACs. Baseline methods specifically tailored for efficient domain adaptation (*e.g.*, DDA and REDA) outperform other baselines like MSDNet, and ResNets. While comparing with DDA and REDA (Li et al., 2021b) over the same budget range, **AnyDA** performs significantly better, specifically for low-budget networks. E.g., we outperform DDA by almost **8**$\%$ at a budget of $0.2 \times 10^9$ MACs, while REDA by $6.9\%$. **AnyDA**'s performance, unlike others, does not experience a drastic drop when the available budget is decreased. This behavior clearly shows that our bootstrapped recursive distillation is not only able to train robust low-capacity models, but also ensures simultaneous domain alignment.

**DomainNet.** On the large-scale DomainNet dataset (ref Figure 3b) also, **AnyDA** outperforms DDA and other baselines. Similar to Office-31, **AnyDA** does especially well at very low computation

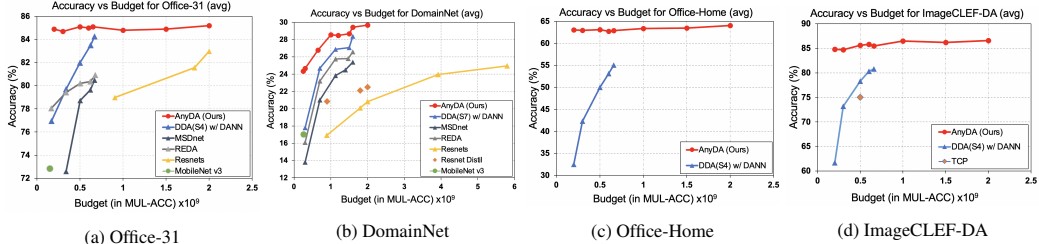

(a) Office-31  (b) DomainNet  (c) Office-Home  (d) ImageCLEF-DA

Figure 3: **Performance on Office-31, DomainNet, Office-Home and ImageCLEF-DA .** The plots compare the average accuracy vs budget curves of our proposed approach, **AnyDA** with different baselines. **AnyDA** clearly outperforms the compared methods in all the budget configurations on all the datasets. Specifically in the low budget settings, we observe the maximum improvement showing the effectiveness of **AnyDA** in learning domain adaptive networks for very low-resource applications. Best viewed in color.

Table 1: **Performance on DomainNet**. We show the task-wise accuracy on the DomainNet dataset for 30 adaptation tasks, where in each sub-table, the columns are the source domains and the rows are the target domains. Our proposed **AnyDA** outperforms DDA (w/ DANN) at both the compared budgets.

| | ResNet-50: $0.7 \times 10^9$ macs | | | | | | | | ResNet-152: $1.6 \times 10^9$ macs | | | | | | |
|---|---|---|---|---|---|---|---|---|---|---|---|---|---|---|---|
| DDA(S4) | clp | inf | pnt | qdr | rel | skt | Avg | DDA(S7) | clp | inf | pnt | qdr | rel | skt | Avg |
| clp | - | 15.5 | 33.8 | 18.5 | 47.0 | 36.2 | 30.2 | clp | - | 16.8 | 36.3 | 20.7 | 51.3 | 39.0 | 32.8 |
| inf | 28.2 | - | 26.0 | 8.4 | 38.0 | 21.1 | 24.3 | inf | 29.4 | - | 28.1 | 10.0 | 43.8 | 23.0 | 26.9 |
| pnt | 37.6 | 15.9 | - | 8.9 | 48.1 | 31.8 | 28.5 | pnt | 40.4 | 17.2 | - | 10.9 | 52.1 | 33.9 | 30.9 |
| qdr | 21.5 | 2.8 | 7.4 | - | 15.1 | 13.0 | 11.9 | qdr | 21.2 | 3.0 | 8.4 | - | 18.6 | 14.1 | 13.1 |
| rel | 43.4 | 18.1 | 41.8 | 9.4 | - | 30.7 | 28.7 | rel | 46.0 | 18.7 | 44.9 | 11.8 | - | 33.9 | 31.1 |
| skt | 49.5 | 16.4 | 36.6 | 17.9 | 47.0 | - | 33.5 | skt | 51.1 | 17.3 | 40.0 | 20.5 | 50.9 | - | 36.0 |
| Avg | 36.0 | 13.7 | 29.1 | 12.6 | 39.0 | 26.6 | 26.2 | Avg | 37.6 | 14.6 | 31.5 | 14.8 | 43.3 | 28.8 | 28.4 |
| | ResNet-50: $0.7 \times 10^9$ macs | | | | | | | | ResNet-50: $1.6 \times 10^9$ macs | | | | | | |
| AnyDA | clp | inf | pnt | qdr | rel | skt | Avg | AnyDA | clp | inf | pnt | qdr | rel | skt | Avg |
| clp | - | 33.2 | 38.4 | 21.2 | 48.8 | 49.3 | 38.2 | clp | - | 33.4 | 40.6 | 23.1 | 51.7 | 53.1 | 40.4 |
| inf | 13.1 | - | 14.2 | 2.5 | 17.1 | 13.3 | 12.0 | inf | 18.2 | - | 17.1 | 3.8 | 21.3 | 17.6 | 15.6 |
| pnt | 32.8 | 31.0 | - | 6.6 | 46.4 | 38.6 | 31.1 | pnt | 37.8 | 33.2 | - | 8.2 | 51.2 | 42.9 | 34.7 |
| qdr | 13.7 | 5.6 | 6.6 | - | 8.1 | 16.4 | 10.1 | qdr | 12.2 | 4.8 | 4.8 | - | 7.6 | 14.2 | 8.7 |
| rel | 47.9 | 44.7 | 51.5 | 13.7 | - | 48.5 | 41.3 | rel | 51.9 | 47.2 | 54.2 | 16.2 | - | 52.9 | 44.5 |
| skt | 35.4 | 24.3 | 32.3 | 13.4 | 34.7 | - | 28.0 | skt | 41.8 | 26.1 | 38.9 | 17.5 | 39.6 | - | 32.8 |
| Avg | 28.6 | 27.8 | 28.6 | 11.5 | 31.0 | 33.2 | **26.8** | Avg | 32.4 | 28.9 | 31.1 | 13.8 | 34.3 | 36.1 | **29.4** |

budgets. Our average accuracy at the lowest budget of $0.25 \times 10^9$ MACs is 24.4% which is about **6.6**% more than DDA at a slightly higher lowest budget of $0.3 \times 10^9$ MACs. It is worth noting that DDA used 7 layered ResNet-152 model in its S7 variant as the backbone while **AnyDA** uses a lower complexity backbone, ResNet-50 with comparatively weak representation learning ability. Similarly, when compared to the next best approach (MobileNet-v3) available at the same lowest budget as ours, the performance is better by 7.4%. At the highest budget ($1.6 \times 10^9$ MACs) available for DDA, **AnyDA** achieves **1**% improved performance of 29.4% over it. The performance improvement is continued as more computation in the anytime framework is available and reaches 29.7% at our highest budget of $2.0 \times 10^9$ MACs. To better illustrate that **AnyDA** can effectively enhance domain alignment at the highest budgets of DDA S4 and S7 architectures, we provide a finegrained comparative analysis on all 30 tasks of DomainNet in Table 1. It can be seen that even though **AnyDA** uses a low capacity backbone (ResNet-50 vis-a-vis ResNet-152), our approach can exploit the goodness of the higher subnetworks for domain alignment in a resource constrained scenario.

**Office-Home.** In Office-Home (ref Figure 3c), we again outperform DDA for all the budgets. Our average accuracy at the lowest budget of $0.25 \times 10^9$ MACs outperforms DDA by **30**%, while by 7.9% at $0.7 \times 10^9$ MACs of budget, and achieves a best accuracy of 64% at $2.0 \times 10^9$ MACs.

**ImageCLEF-DA.** Our performance in ImageCLEF-DA (ref Figure 3d) is also in accordance with the trend in the rest of the datasets. We outperform DDA in all budgets significantly. At the lowest budget of $0.25 \times 10^9$ MACs **AnyDA** is better by an absolute margin of 23.1%, while by 4.7% at $0.7 \times 10^9$ MACs. As in other datasets, the performance of **AnyDA** is minimally hampered with decreasing computation budgets, showing the efficacy of our approach for low-capacity networks.

## 4.2 COMPARISON WITH SLIMDA

We also compared our performance with SlimDA (Meng et al., 2022), most close to ours. Due to unavailability of the SlimDA code at the time of submission, we could compare the performance on

Table 2: **Comparision with SlimDA.** We show average accuracy for Office-31, Office-Home and ImageCLEF-DA. **AnyDA** consistently shows better performance compared to SlimDA on 4 different budgets.

| | Office-31 | | | | Office-Home | | | | ImageCLEF-DA | | | |
|---|---|---|---|---|---|---|---|---|---|---|---|---|
| Flops | $1\times$ | $\frac{1}{2}\times$ | $\frac{1}{4}\times$ | $\frac{1}{10}\times$ | $1\times$ | $\frac{1}{2}\times$ | $\frac{1}{4}\times$ | $\frac{1}{8}\times$ | $1\times$ | $\frac{1}{2}\times$ | $\frac{1}{4}\times$ | $\frac{1}{8}\times$ |
| SlimDA (Meng et al., 2022) | 86.3 | 87.7 | 87.4 | 87.5 | 68.4 | 68.0 | 67.8 | 67.4 | 88.9 | 88.7 | 88.8 | 88.4 |
| **AnyDA** | 88.2 | 87.9 | 87.8 | 87.2 | 68.7 | 68.1 | 68.1 | 67.5 | 90.4 | 89.2 | 89.5 | 89.2 |

the three datasets Office-31, Office-Home and ImageCLEF-DA on which the reported results were obtained. SlimDA builds on top of more recent and stronger domain adaptation approach Sym-Net (Zhang et al., 2019b) by adding stochastic ensemble distillation with it. For a fair comparison, we also adopted **AnyDA** with the SymNet architecture by simply equipping the ResNet-50 back-bone of SymNet with switchable depth, width and input resolution and employing the bootstrapped recursive distillation therein without needing to change the original biclassifier design of SymNet. Table 2, shows **AnyDA** is better than SlimDA in all datasets across various computation budgets while being competitive at lowest compared budget of $\frac{1}{10}\times$ on Office-31. On the other hand, the full budget comparison shows an absolute improvement of $1.9\%$ on Office-31 dataset. This experiment shows the easy adaptability of **AnyDA** with other sophisticated domain adaptation frameworks towards efficient domain adaptation under budget constraints.

We also combine SymNet (Zhang et al., 2019b), with anytime prediction (US-Net (Yu & Huang, 2019) with slimmable width, depth and resolution as ours) naively in three possible baseline settings on Office-31 and Office-Home to verify the performance of directly combining existing domain adaptation and anytime prediction techniques. Table 3 in Appendix A shows that a direct combination of these two existing methods fails miserably for the lower budget subnets even with additional pseudo-labeling added on top of it for harnessing categorical information from the target domain. This clearly corroborates the importance of the proposed components in **AnyDA** towards learning a robust network executable at different computation budget as well as alleviating the domain shift.

## 4.3 ABLATION STUDIES

We perform ablation studies on Office-31 dataset (unless otherwise specified) to test the effectiveness of different components of **AnyDA** and different variations of bootstrapped recursive distillation.

**Training Supernet with Only Source Data.** This ablation tests the naive scenario of employing a single high capacity network for anytime domain adaptation with a goal to set a lower bound of the performance. We train only the supernet (a ResNet-50 model only) with source data and test it in all the computation budgets in the target domain. Figure 4a shows that the performance is poor as expected. Especially, the lower subnets suffer the most with fluctuations in the performance.

**Effect of Teacher Selection for Distillation.** We verify the advantage of recursive distillation by comparing with two other variants. In the first variation, the recursive distillation loss encourages all student subnets including the minnet to mimic the highest capacity teacher supernet by modifying the first term in Eqn. 2 from $\mathrm{H}(p_{t1}^{stu}, p_{tavg}^{tea})$ to $\mathrm{H}(p_{t1}^{stu}, p_{tn}^{tea})$. From Figure 4b, we see consistent performance drop across all budgets by around $1\%$ compared to the proposed approach where the minnet learns from the average of the teacher subnets. We conjecture that the progressively lower capacity and thus continually increasing gap of the student subnets with the teacher supernet can cause convergence hardship. In the second variation, we study the effect of mimicking the average of the soft logits of all the teacher subnets, supernet and minnet. Figure 4c shows that the performance drops by around $2\%$ consistently across all computation budgets showing that our proposed recursive distillation strategy makes better use of the knowledge from multiple teacher subnets.

**Training by Non-Recursive Distillation.** Here, we experiment with non-recursive distillation where each student subnet learns from its companion subnet in the teacher. This is done by encouraging each $p_{ti}^{stu}$ to mimic the corresponding soft logit $p_{ti}^{tea}$ from the teacher resulting in $\mathcal{L}_{rd}(x_t)$ to be $\sum_{i=1}^{n} \mathrm{H}(p_{ti}^{stu}, p_{ti}^{tea})$ in Eqn. 2. As seen in Figure 4d, this results in a drastic performance drop especially for the lower subnets showing the importance of the flow of information progressively from the top of the ladder to the bottom. Interestingly, the importance of recursive distillation can be appreciated from the fact that without it, the performance drops even below the naive approach of training a single supernet with only source data (compare with Figure 4a). Another variant of non-recursive learning is not to learn from the bootstrapped teacher, instead try to mimic its nearest

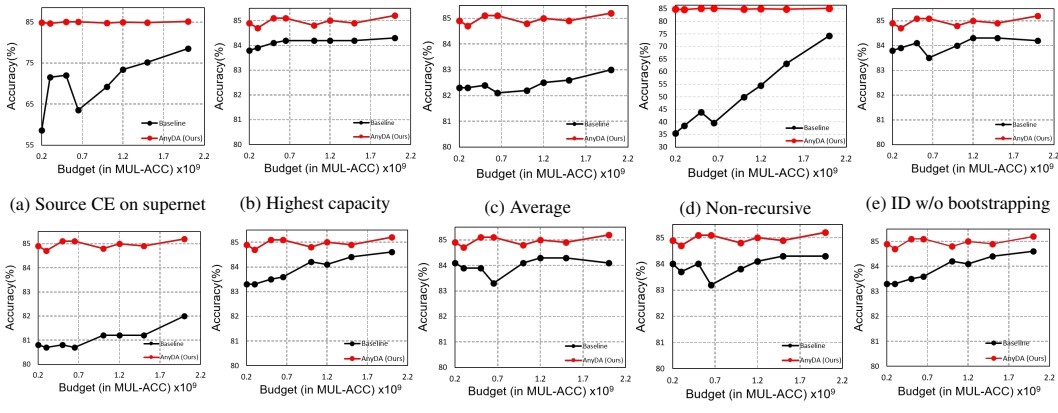

Figure 4: **Ablation Studies on Office-31.** The *red* line shows the accuracy curve for **AnyDA** while the *black* line corresponds to that of the baseline with specific ablation setting. **AnyDA** outperforms the corresponding baseline in all settings showing its effectiveness in anytime domain adaptation. Best viewed in color.

superior subnet from the student itself. Such inplace distillation (ID) works well with no distribution shift (Yu & Huang, 2019). However such a strategy (ID w/o bootstapping) naturally fails for the current task as it does not use the information from supervised source data warmup. Thus, Figure 4e shows significant performance drop across all budgets. A fairer comparison would be to employ the teacher for inference which starting with source data warmup updates itself by exponential moving average of the student weights. However, Figure 4f shows that even such an inplace distillation with bootstapping (ID w/ bootstapping) does not also work well for anytime domain adaptation.

**Ablation on Distillation Loss.** We investigate the effectiveness of cross-entropy loss (ref Eqn. 2) by replacing it with KL Divergence loss traditionally used in knowledge distillation. Figure 4g shows that training with cross-entropy loss surpasses the performance of training with KL Divergence loss by a high margin especially for the lower subnets (around 1.6 % average accuracy drop in minnet).

**Ablation on Batch Normalization.** Using batch normalization is crucial for better convergence of both domain adaptation (Chang et al., 2019) as well as slimmable networks (Yu et al., 2019b). We ran a variant of **AnyDA** where switchable batch normalization layers were used for the different subnets but these were not different for the source and target domain data. The performance degraded by roughly 1% across all computation budgets on average (ref Figure 4h). We also tried using a single domain-specific batch-normalization for all the subnets but the testing accuracy was extremely poor. These results help us conclude the need for both domain as well as subnet specific batch normalization to achieve the twin goals of domain generalization and computation efficiency.

**Need for Explicit Domain Discriminator.** Using domain discriminators to confuse domains has been a popular choice for domain adaptation. However, our experiments in Figure 4i show that in anytime setting, adding an explicit domain discriminator (Ganin & Lempitsky, 2015) may not offer any advantage, rather the presence of many subnetworks and as a result, multi-scale features can deteriorate the discriminability of the framework.

**Role of Pseudo-Labeling.** We remove the pseudo-label loss $\mathcal{L}_{pl}$ and observe that **AnyDA** without target pseudo-labels decreases the performance by more than 1% on average (ref Figure 4j) indicating the advantage of pseudo-labeling in enhancing discriminability by successfully leveraging label information from target domain. Additional results and analysis are included in the Appendix G.

## 5 CONCLUSION

In this paper, we introduce a novel approach for anytime domain adaptation by considering domain alignment with switchable depth, width and input resolutions to achieve accuracy-efficiency trade-offs in the target domain for different resource constraints. In particular, we adopt a teacher-student framework with bootstrapped recursive distillation to bring the target features close to the source and also to transfer the learned knowledge for efficient inference. We also leverage self-supervision via pseudo-labeling on the student supernets to ensure a discriminative latent space for the unlabelled target images. We demonstrate the effectiveness of our proposed approach on four benchmark datasets, outperforming several competing methods.

ACKNOWLEDGEMENTS

This work was partially supported by Google India Research Award 2021. OC and AD sincerely acknowledge the travel reward partially supporting their travel for presenting this work. We also acknowledge support from the MIT-IBM Watson AI Lab, which—through MIT Satori Supercomputer—contributed the computational resources necessary to conduct the experiments in this paper.

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

# A    COMPARISON WITH NAIVE COMBINATION OF ANYTIME PREDICTION AND DOMAIN ADAPTATION

In Table 3, we combine one existing state-of-the-art domain adaptation (DA) method, namely SymNet (Zhang et al., 2019b), with anytime prediction (AP) (US-Net (Yu & Huang, 2019) with slimmable width, depth and resolution as ours) naively in three settings: (1) Domain adaptation + Anytime prediction (inference only): in this experiment, we first train a network using the SymNet domain adaptation method. Once trained, we used the adapted model directly for anytime inference to obtain its performance at various computation budgets on the target domain. (2) Domain adaptation + Anytime prediction (w/ Pseudo-labeling): in this experiment, similar to point (1) above, we obtain an adapted network using the SymNet domain adaptation technique on source and target data. After that, we further train the network in an anytime fashion using US-Net on the target data using self-supervision through pseudo-labels (PL). Note that since the target data is unlabeled, we need to use pseudo-labels to train on them using existing anytime networks. (3) Anytime prediction + Domain Adaptation: in this experiment, we first train a network for anytime prediction on the labeled source data. Then, we train it using the SymNet domain adaptation approach. We compare the performance of **AnyDA** using the same SymNet backbone network for a fair comparison.

Table 3: **Comparison of `AnyDA` with naive combinations of anytime prediction and domain adaptation** on Office-31 and Office-Home. DA: Domain Adaptation, AP: Anytime Prediction, PL: Pseudo-Labeling. We use SymNet backbone for all the experiments.

| | **Office-31** | | | | | **Office-Home** | | | | |
|---|---|---|---|---|---|---|---|---|---|---|
| Flops | $1\times$ | $\frac{1}{2}\times$ | $\frac{1}{4}\times$ | $\frac{1}{10}\times$ | Avg | $1\times$ | $\frac{1}{2}\times$ | $\frac{1}{4}\times$ | $\frac{1}{10}\times$ | Avg |
| DA + AP (inference only) | 86.4 | 35.7 | 31.8 | 30.0 | 46.0 | 62.8 | 24.5 | 22.9 | 20.8 | 32.8 |
| DA + AP (w/ PL) | 87.9 | 43.6 | 38.5 | 37.7 | 51.9 | 62.7 | 31.9 | 28.4 | 27.6 | 37.7 |
| AP + DA | 73.6 | 46.3 | 43.2 | 40.0 | 50.8 | 56.5 | 47.0 | 35.6 | 32.2 | 42.8 |
| **AnyDA** | 88.2 | 87.9 | 87.8 | 87.2 | 87.8 | 68.7 | 68.1 | 68.1 | 67.5 | 68.1 |

We have the following observations from Table 3, (1) DA + AP (inference only): simply performing anytime inference on a domain adapted model performs poorly as compared to **AnyDA** even when using the full network with the highest budget (1.8% lower in Office-31, 5.9% lower in Office-Home) and fails miserably for the lower budget subnets (e.g. 57.2% lower in Office-31, 46.7% lower in Office-Home in the lowest budget configuration); (2) DA + AP (w/ PL): as can be seen, the performance improves as compared to point (1) because of the additional anytime training on the target data. But, the the huge drop in performance from the highest budget network to the lower budget subnets is still significant (e.g. 49.5% and 39.9% lower than **AnyDA** in the lowest budget network); (3) While this variation performs better at lower budgets, the performance at the highest budget is lower than the other two variants, showing that anytime training without exploiting target data can give a poor initialization for anytime domain adaptation. Our approach outperforms AP+DA variant by 14.6% and 12.2% at the highest budget, while significantly outperforming it by more than 30% at the lower budgets for the Office-31 dataset. Similar observations can be made in the Office-Home dataset as well. To summarize, all the findings above corroborate the importance of the proposed components in **AnyDA** towards learning a robust network executable at different computation budget as well as alleviating the domain shift.

## B  PERFORMANCE USING LARGER BACKBONE

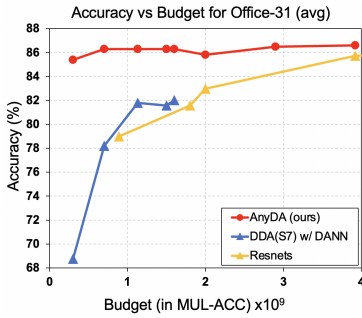

In Figure 5, we perform additional experiments using ResNet-101 on Office-31 dataset in order to show the effectiveness of our approach in larger backbones. We train DDA using the largest available supported backbone of S7. As can be seen, **AnyDA** achieves the best average accuracy of 86.4% at a budget of $3.9 \times 10^9$ MACs. At the highest comparable budget with DDA of $1.6 \times 10^9$ MACs, **AnyDA** significantly outperforms by 4.2%. **AnyDA**'s performance, unlike others, does not experience a drastic drop when the available budget is decreased. The improvement obtained at the lower budgets is even significantly more, e.g., at the corresponding lowest comparable budgets, **AnyDA** outperforms DDA by 16.2% at $0.3 \times 10^9$ MACs and ResNet by 5.9% at $0.9 \times 10^9$ MACs. This behavior clearly shows that our bootstrapped recursive distillation is not only able to train robust low-capacity models, but also ensures simultaneous domain alignment with larger backbones.

Figure 5: **Performance on Office-31.** Plots compare the average accuracy vs budget curves for **AnyDA** using ResNet-101 backbone, with two other baselines. Best viewed in color.

## C  FEATURE VISUALIZATION UNDER DIFFERENT BUDGETS

Figure 6 shows the t-SNE visualizations on four adaptation tasks (A→W, D→A, A→D, and D→W) from the Office-31 dataset. The figure shows the clustering of the target features at various computation budget subnets from the supernet in the left to the minnet in the right for **AnyDA**. As can be seen from Figure 6, the clustering is fairly consistent and discriminative across the subnets till the network with budget $0.6 x 10^9$ MACs while slightly slackening for the very low budgets (after 5th column), showing the effectiveness of the bootstrapped recursive distillation in learning discriminative feature space at different budget configurations.

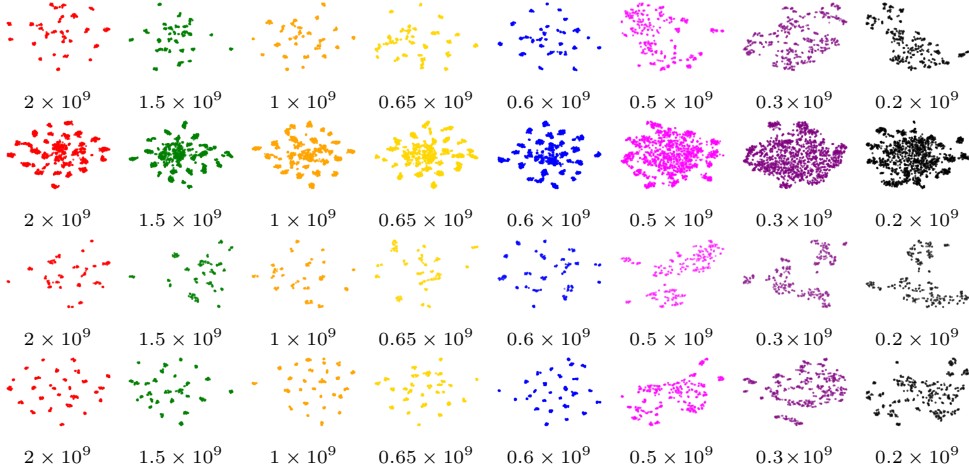

Figure 6: **Feature Visualization using t-SNE.** Figure shows the t-SNE visualizations of the target features on four adaptation tasks (A→W, D→A, A→D, and D→W, from top to bottom, respectively) from the Office-31 dataset. The computation budgets in MACs are mentioned under the plots, with supernet at the extreme left, while minnet at the extreme right. Best viewed in color.

## D  DISCUSSION ON UTILITY OF ANYTIME DOMAIN ADAPTATION

The focus of our work is on learning a single network which is domain invariant and can be executed at different computational budgets (MACs) in multiple devices with different budget requirements. Training a single network with the ability to be executed at different budgets is more feasible and efficient than training separate networks of corresponding budget configurations. In many practical applications, once the model is trained, it is an extremely important to perform inference many times (without retraining) due to highly dynamic deployment environments (train once but inference

many). Additionally, computation cost of training a network for `AnyDA` is more than that of a conventional network due to joint training of all the subnets. However, we follow *sandwich rule* to reduce the training-time computation overhead by almost 4 times as compared to forward-passing through all the subnets. Focussing on both training-time as well as inference-time efficiency is an interesting research topic, which would be an exciting future work.

## E   DATASET DETAILS

We evaluate the performance of our approach on four benchmark datasets, namely, (1) Office-31 (Saenko et al., 2010), (2) Office-Home (Venkateswara et al., 2017), (3) DomainNet (Peng et al., 2019) and (4) ImageCLEF-DA (Long et al., 2017). Below we provide them in details.

**Office-31.** This dataset contains $4,110$ images distributed among 31 different categories and collected from three different domains: Amazon (**A**), Webcam (**W**) and Dslr (**D**), resulting in 6 transfer tasks. Amazon images are collected from merchant websites and possess a clear white background. Webcam images are captured using a web-cam and are of lower resolution. Dslr images are low-noise high resolution images. The dataset is imbalanced across domains with $2,817$ images belonging to Amazon, 795 images to Webcam, and 498 images to Dslr, making Amazon a larger domain as compared to Webcam and Dslr. The dataset is publicly available to download at:
`https://people.eecs.berkeley.edu/~jhoffman/domainadapt/#datasets_code.`

**Office-Home.** This dataset contains $15,588$ images distributed among 65 different classes and collected from four different domains: Art (**Ar**), Clipart (**Cl**), Product (**Pr**), and RealWorld (**Rw**), resulting in 12 transfer tasks. The Art domain has images of paintings, sketches and artistic depictions. The Clipart domain consists of clipart images, while Product domain has clear background images. The RealWorld domain has regular images captured with a camera. The dataset is split across domains with 2427 images belonging to Art, 4365 images to Clipart, 4439 images to Product, and 4347 images to RealWorld. The dataset is publicly available to download at:
`http://hemanthdv.org/OfficeHome-Dataset/.`

**DomainNet.** This dataset is one of the largest domain adaptation benchmark datasets available containing around 0.6 million images. The images are distributed among 345 different categories and are collected from six different domains: Infograph (**inf**), Quickdraw (**qdr**), Real (**rel**), Sketch (**skt**), Clipart (**clp**), and Painting (**pnt**), resulting in 30 transfer tasks. We use the cleaned version of the dataset which is split across the domains with $51,605$ images belonging to Infograph, $172,500$ images to Quickdraw, $172,947$ images to Real, $69,128$ images to Sketch, $48,129$ images to Clipart, and $72,266$ images to Painting. The dataset is publicly available to download at:
`http://ai.bu.edu/M3SDA/.`

**ImageCLEF-DA.** The ImageCLEF-DA dataset is a benchmark dataset used in the ImageCLEF domain adaptation challenge of 2014. Following the standard practise, we carried out our experiments considering the three domains, Caltech-256 (**C**), ImageNet ILSVRC 2012 (**I**) and Pascal VOC 2012 (**P**). There are a total of 12 categories in each domain with each category having around 50 images. The dataset is publicly available at:
`http://imageclef.org/2014/adaptation/.`

## F   IMPLEMENTATION DETAILS

The pseudo-code for the training of `AnyDA` is shown in Algorithm 1.

**Training Details.** For Eqn. 1, we use $\tau_{stu} = 0.1$ and $\tau_{tea} = 0.04$. We use a momentum hyperparameter value of $\lambda = 0.96$. In Eqn. 3, a threshold value of $\tau_{pl} = 0.9$ was used for Office-31 and Office-Home dataset, while $\tau_{pl} = 0.4$ for the DomainNet dataset. While in Eqn. 4, we use $\lambda_{cls} = 1, 15, 64$, $\lambda_{rd} = 1, 1, 0.5$ for Office-31, Office-Home and DomainNet, respectively, $\lambda_{pl} = 0.1$ for all the datasets. We perform warm-up using source data for $100, 100, 30$ epochs for Office-31, Office-Home and DomainNet, respectively. The proposed approach `AnyDA` is trained

---

**Algorithm 1** The training pseudocode for **AnyDA**

---

**Data:** source data $\mathcal{D}_s$ and target data $\mathcal{D}_t$, hyperparameters
**Networks:** Student network $\mathcal{G}^{\theta_{stu}*} = \{s_1^{stu}, s_2^{stu}, ..., s_n^{stu}\}$ and Teacher network $\mathcal{G}^{\theta_{tea}*} = \{s_1^{tea}, s_2^{tea}, ..., s_n^{tea}\}$

1: warmup the networks $\mathcal{G}^{\theta_{stu}*}$ using source data $\mathcal{D}_s$
2: $\theta_{tea} = \theta_{stu}$
   # load mini-batch and iterate over dataloader
3: **for** $s$, $t$ in loader **do**
4:    obtain predictions $p_t i^{tea}$ for all $s_i^{tea}$ in $\mathcal{G}^{\theta_{tea}*}$
5:    compute $p_{tavg}^{tea}$
     # Sandwich rule
6:    stu_subnets ← minnet, two randomly sampled subnets, supernet from $\mathcal{G}^{\theta_{stu}*}$
     # Iterate over the student subnets
7:    **for** $s_i^{stu}$ **in** stu_subnets **do**
8:       obtain predictions $p_{si}^{stu}, p_{ti}^{stu}$
9:       compute cross-entropy loss for $p_{si}^{stu}$
10:      compute recursive distillation loss for $p_{ti}^{stu}$ using H(.)
11:      compute pseudo-label loss for $p_{ti}^{stu}$, if $s_i^{stu}$ is supernet
12:    **end for**
13:    compute $\mathcal{L}_{cls}, \mathcal{L}_{rd}, \mathcal{L}_{pl}$
14:    optimize $\mathcal{G}^{\theta_{stu}}$ using gradient descent
     # Update teacher network using ema
15:    update teacher network: $\theta_{tea} \leftarrow \lambda\theta_{tea} + (1 - \lambda)\theta_{stu}$
16: **end for**
17: **def** H($p_s, p_t$):
     # Stop gradient
18:    $p_t = p_t$.detach()
19:    return - $(p_t * \log(p_s))$.sum(dim=1).mean()

---

for $30, 100, 20$ epochs, respectively. Different input configurations are handled by our network as: *depth*: for a depth of $d$, we only execute first $d\times100\%$ blocks in each layer. E.g. layer-1 of ResNet-50 has 3 blocks, so for $d$=0.5, we execute only the first $\lfloor 3\times0.5 \rfloor$=1 block of layer-1. *width*: Similarly, for a width of $w$, we only execute first $w\times100\%$ filters/weights for all layers. *resolution*: we consider $\mathcal{R} = \{224, 192, 160, 128\}$ which are compatible with convolution kernel size, padding and strides of ResNet-50 network. Following (Yu & Huang, 2019), we use a single linear layer as our classifier. Specifically, for a linear layer of shape $d \times n$ ($n$ = # classes), given an width scale of 0.9, the subset weights of shape $0.9d \times n$ is used. We use a per-gpu batch size of $64$ (32 source + 32 target) for all the experiments. We use a learning rate of $2e$-4 for Office-31 and Office-Home, while $3e$-5 for DomainNet. We follow cosine annealing to update the learning rate. We report the average classification accuracy. We additionally use information maximization on target data as a regularizer. All the experiments were performed using 4 NVIDIA Tesla V100 GPUs.

**Budget Calculation.** In Table 4, we show the budget values for all the 16 configuration tuples. Specific budgets at which our network is executed during inference are not pre-set, instead only a range of budgets (minimum and maximum) for defining subnets during training is used in our current work. As can be seen, we obtain 8 intervals of equal size from the 16 configurations. After training, our trained model (the student network) is executable at various budget configurations. The goal is to find the best configuration under a particular resource constraint. We achieve this by using a query table. For example, in ResNet-50, we sample network width from $\{1, 0.9\}$, network depth from $\{1, 0.5\}$ and sample input resolution from $\{224, 192, 160, 128\}$. We test all these width-depth-resolution configurations on a validation set and choose the best one under a given budget at inference. Since there is no re-training, the whole process is once for all. For handling budgets that fall outside of this range, one can train a supernet with a very large backbone or by considering extreme low values of width, depth and resolution: we leave this as an interesting future work.

Table 4: **Budget Calculation.** We show the details of all the configuration tuples and their corresponding budget values. The third column groups the subnetworks having computation budget within a defined range. We divide the whole computation budget of ResNet-50 into equally spaced 8 ranges of size $0.25 \times 10^9$ macs. If multiple subnetworks come under this range (e.g., two subnetworks denoted by $< 1 \times 0.9 \times 192 >$ and $< 0.5 \times 1 \times 224 >$ in the first column come in the range of $1 \times 10^9$ to $1.25 \times 10^9$), then we report the result given by the best performing subnetwork.

| ResNet-50 | | |
|---|---|---|
| $\mathcal{R}$ x $\mathcal{W}$ x $\mathcal{D}$ | **Budget** (macs $\times 10^9$) | **Budget Range** (macs $\times 10^9$) |
| 224 x 1.0 x 1.0 | 2.00 | (1.75,2.00] |
| 224 x 0.9 x 1.0 | 1.65 | (1.50,1.75] |
| 192 x 1.0 x 1.0 | 1.50 | (1.25-1.50] |
| 192 x 0.9 x 1.0 | 1.20 | (1.00-1.25] |
| 224 x 1.0 x 0.5 | 1.05 | |
| 160 x 1.0 x 1.0 | 1.00 | (0.75-1.00] |
| 224 x 0.9 x 0.5 | 0.85 | |
| 160 x 0.9 x 1.0 | 0.85 | |
| 192 x 1.0 x 0.5 | 0.75 | (0.50-0.75] |
| 128 x 1.0 x 1.0 | 0.65 | |
| 192 x 0.9 x 0.5 | 0.60 | |
| 128 x 0.9 x 1.0 | 0.55 | |
| 160 x 1.0 x 0.5 | 0.50 | (0.25, 0.50] |
| 160 x 0.9 x 0.5 | 0.40 | |
| 128 x 1.0 x 0.5 | 0.30 | |
| 128 x 0.9 x 0.5 | 0.20 | (0.00, 0.25] |

## G    ADDITIONAL EXPERIMENTS

**Effect of Hyperparameters.** We study the effect of the coefficient hyperparameters $\lambda_{rd}$ and $\lambda_{pl}$. For Office-31, we consider values of $\lambda_{rd}$ and $\lambda_{pl}$ an order above and below the best value used in our experiments and show the obtained results in Figure 7. We observe that $\lambda_{rd} = 1.0$ and $\lambda_{pl} = 0.1$ gives the best performance.

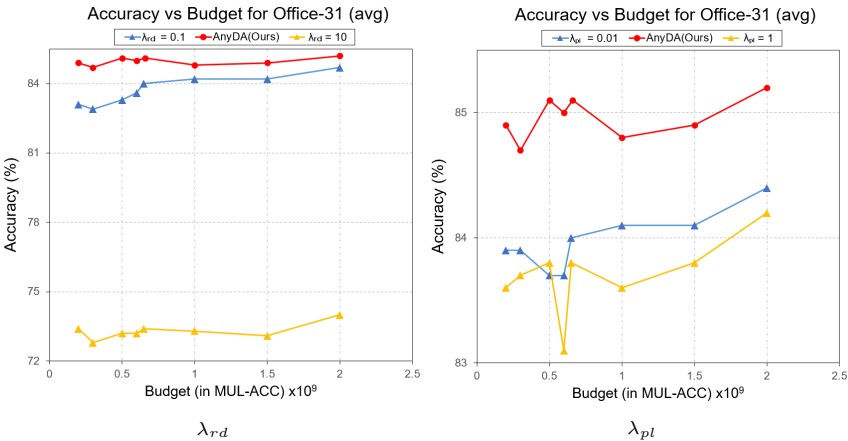

$\lambda_{rd}$          $\lambda_{pl}$

Figure 7: **Effect of hyperparameters.** The plots show average accuracy vs budget by varying the hyperparameter values $\lambda_{rd}$ and $\lambda_{pl}$ (ref. Eqn. 5 in main paper) for the Office-31 dataset.

**Ablation Studies on OfficeHome.** Similar to the main paper, we perform ablation studies on the OfficeHome dataset, and report all the findings in Figure 8. For all the cases, **AnyDA** outperforms the corresponding ablation setup, which affirms the importance of the proposed bootstrapped recursive distillation method along with the other components for anytime domain adaptation.

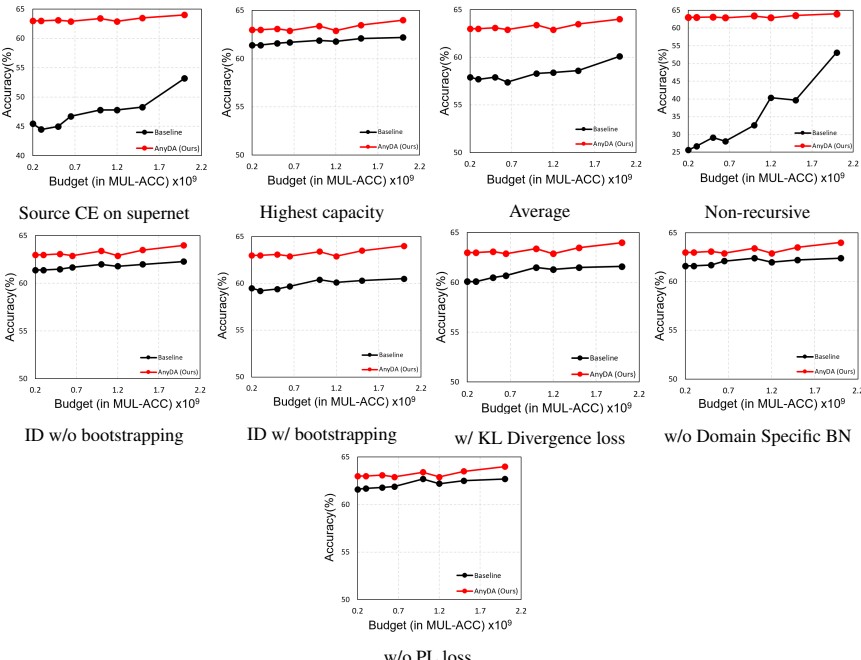

Figure 8: **Ablation Studies.** The plots show average accuracy vs budget curves for different ablation studies performed on Office-Home dataset. The *red* line shows the accuracy curve for `AnyDA` while the *black* line corresponds to that of the baseline with specific ablation setting. Here we observe a similar trend of the individual ablation studies with those for Office-31 dataset. For Office-Home also, `AnyDA` outperforms the corresponding baseline in all settings showing its effectiveness in anytime domain adaptation. Best viewed in color.

**Benefit of Recursive Distillation.** It was shown in (Yang et al., 2020) that decreasing the lower bound of computation budget can result in decrease of the model performance at higher budgets. In this experiment, we tested by running `AnyDA` with increased lower bound of the computation from $0.2 \times 10^9$ macs to $0.5 \times 10^9$ macs on the Office-31 dataset and studied the performance in Figure 9. We observe that the increased budget at lower end of the subnetworks provides a very similar performance with slight improvements. This shows the effectiveness of the recursive distillation method in learning robust representations even if we have very low budget subnetworks which can not drag down the performance of the higher end subnetworks, rather the lower end subnetworks perform almost as good as the higher end ones.

## H  BROADER IMPACT

Our research on anytime domain adaptation can help reduce burden of collecting large-scale supervised data in many real-world applications by transferring knowledge from auxiliary datasets. At the same time, it can have a positive impact on many applications that require customization of a single deep neural network in the target domain to meet the dynamically changing demand. Our research on anytime domain adaptation can also reduce the memory and power consumption, leading to a high impact on the environment as AI systems become more prevalent. Potential negative impacts share many of the pitfalls associated with standard deep learning models such as vulnerability to adversarial attacks and dataset bias, and lack of interpretability, etc.

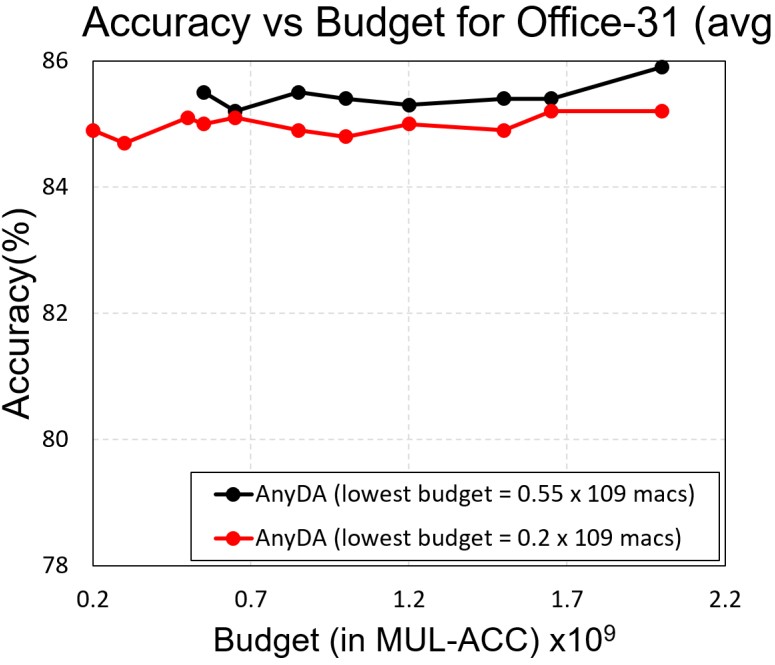

Figure 9: **Effect of increasing budget lower bound.** Plot shows difference in average accuracy vs budget curves by increasing the lower bound of budget from $0.2 \times 10^9$ macs to $0.55 \times 10^9$ macs on Office-31.

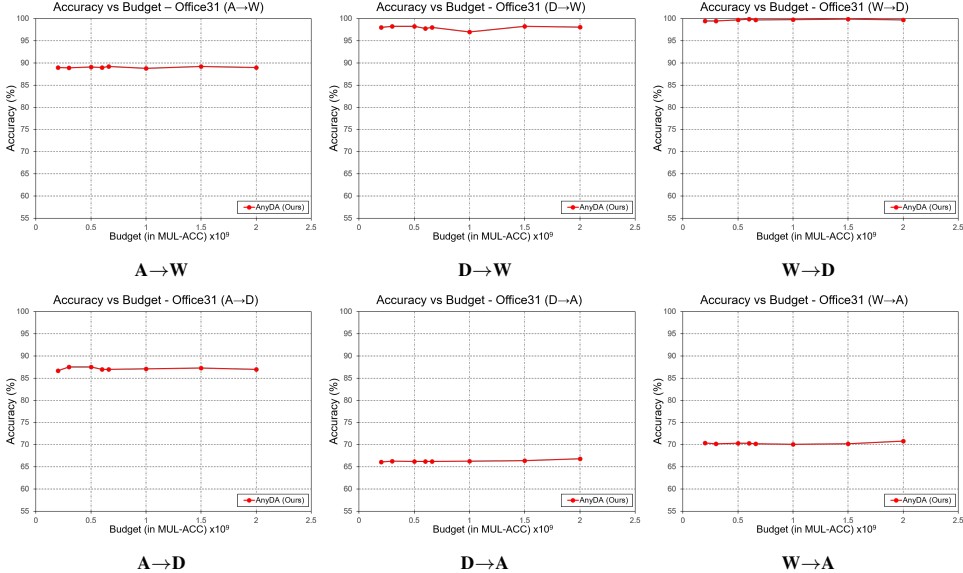

Figure 10: **Performance on Office-31.** The plots 10a to 10f show the accuracy vs budget curves for all the six adaptation tasks of the Office-31 dataset. Best viewed in color.

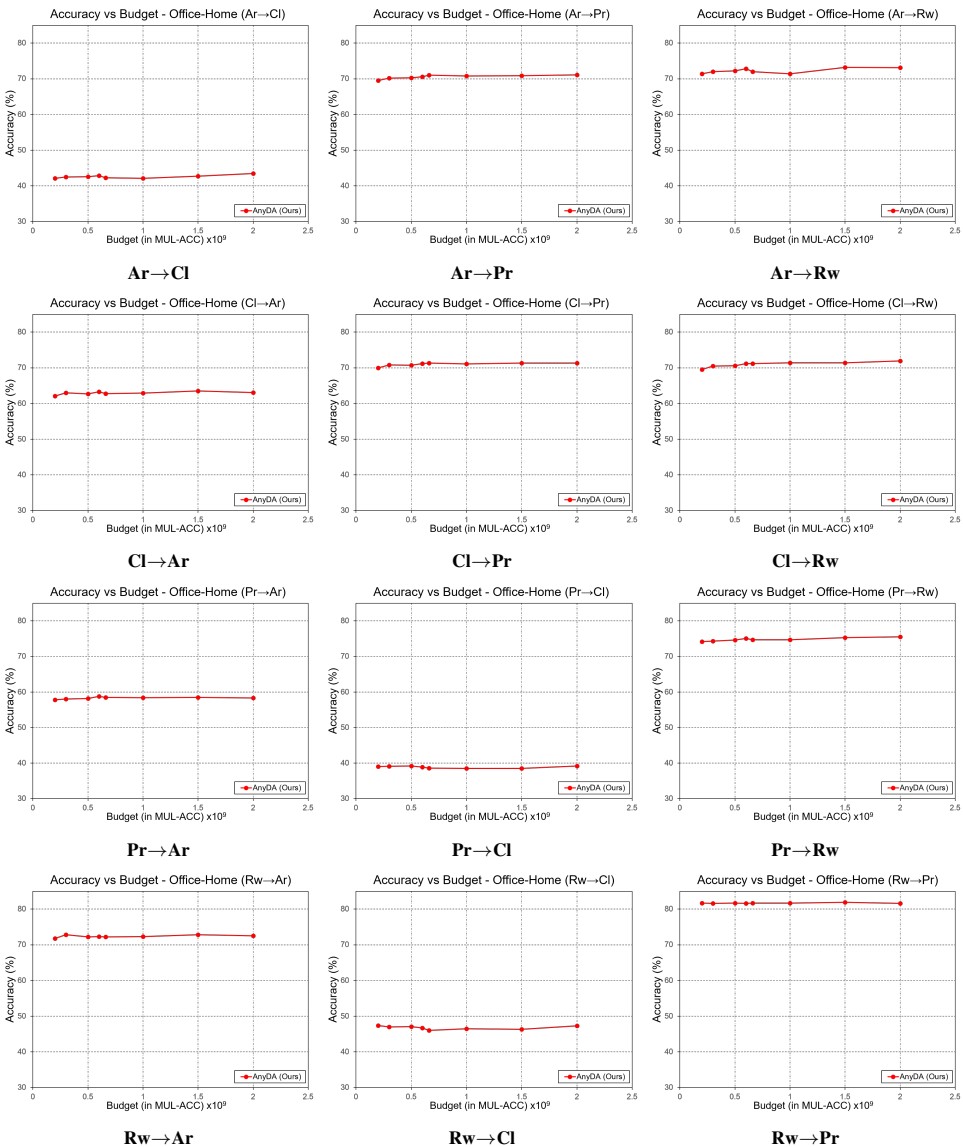

Figure 11: **Performance on Office-Home.** The plots 11a to 11l show the accuracy vs budget curves for all the twelve adaptation tasks of the Office-Home dataset.

Table 5: **Performance on DomainNet**. We show the task-wise accuracy on the DomainNet dataset for 30 adaptation tasks and for all the 8 budget intervals, where in each sub-table, the column-wise domains are the source domain and the row-wise domains are the target domain. The corresponding budget values are given above each of the sub-table. We use ResNet-50 backbone for all our experiments.

$0.2 \times 10^9$ macs

| AnyDA | clp | inf | pnt | qdr | rel | skt | Avg |
|---|---|---|---|---|---|---|---|
| clp | - | 28.9 | 34.0 | 19.8 | 45.5 | 46.3 | 34.9 |
| inf | 11.9 | - | 12.8 | 2.2 | 15.3 | 11.8 | 10.8 |
| pnt | 29.9 | 28.7 | - | 5.6 | 43.6 | 35.7 | 28.7 |
| qdr | 11.6 | 3.2 | 4.8 | - | 5.6 | 14.3 | 7.9 |
| rel | 45.2 | 41.6 | 48.6 | 12.7 | - | 45.1 | 38.6 |
| skt | 31.9 | 21.7 | 28.4 | 12.1 | 32.0 | - | 25.2 |
| Avg | 26.1 | 24.8 | 25.7 | 10.5 | 28.4 | 30.6 | 24.4 |

$0.3 \times 10^9$ macs

| AnyDA | clp | inf | pnt | qdr | rel | skt | Avg |
|---|---|---|---|---|---|---|---|
| clp | - | 29.0 | 34.7 | 19.8 | 45.7 | 46.3 | 35.1 |
| inf | 12.3 | - | 13.0 | 2.3 | 15.9 | 12.4 | 11.2 |
| pnt | 30.6 | 28.6 | - | 5.8 | 44.1 | 36.7 | 29.2 |
| qdr | 11.6 | 4.1 | 4.6 | - | 6.3 | 14.4 | 8.2 |
| rel | 45.7 | 41.7 | 48.8 | 12.4 | - | 45.8 | 38.9 |
| skt | 32.9 | 21.6 | 28.3 | 12.2 | 32.5 | - | 25.5 |
| Avg | 26.6 | 25.0 | 25.9 | 10.5 | 28.9 | 31.1 | 24.7 |

$0.7 \times 10^9$ macs

| AnyDA | clp | inf | pnt | qdr | rel | skt | Avg |
|---|---|---|---|---|---|---|---|
| clp | - | 33.2 | 38.4 | 21.2 | 48.8 | 49.3 | 38.2 |
| inf | 13.1 | - | 14.2 | 2.5 | 17.1 | 13.3 | 12.0 |
| pnt | 32.8 | 31.0 | - | 6.6 | 46.4 | 38.6 | 31.1 |
| qdr | 13.7 | 5.6 | 6.6 | - | 8.1 | 16.4 | 10.1 |
| rel | 47.9 | 44.7 | 51.5 | 13.7 | - | 48.5 | 41.3 |
| skt | 35.4 | 24.3 | 32.3 | 13.4 | 34.7 | - | 28.0 |
| Avg | 28.6 | 27.8 | 28.6 | 11.5 | 31.0 | 33.2 | 26.8 |

$1.0 \times 10^9$ macs

| AnyDA | clp | inf | pnt | qdr | rel | skt | Avg |
|---|---|---|---|---|---|---|---|
| clp | - | 32.9 | 39.1 | 23.0 | 51.1 | 52.9 | 39.8 |
| inf | 16.5 | - | 16.5 | 2.9 | 20.1 | 16.6 | 14.5 |
| pnt | 36.1 | 32.2 | - | 7.4 | 49.9 | 42.6 | 33.6 |
| qdr | 12.2 | 4.4 | 4.8 | - | 7.6 | 14.2 | 8.6 |
| rel | 51.4 | 46.9 | 54.1 | 14.9 | - | 51.9 | 43.8 |
| skt | 39.8 | 25.9 | 35.0 | 15.2 | 38.5 | - | 30.9 |
| Avg | 31.2 | 28.5 | 29.9 | 12.7 | 33.4 | 35.6 | 28.6 |

$1.2 \times 10^9$ macs

| AnyDA | clp | inf | pnt | qdr | rel | skt | Avg |
|---|---|---|---|---|---|---|---|
| clp | - | 32.9 | 39.3 | 23.0 | 51.0 | 52.9 | 39.8 |
| inf | 16.5 | - | 16.4 | 2.9 | 20.1 | 16.6 | 14.5 |
| pnt | 36.1 | 32.0 | - | 7.4 | 49.6 | 42.6 | 33.5 |
| qdr | 12.4 | 4.4 | 4.9 | - | 7.6 | 14.2 | 8.7 |
| rel | 50.3 | 46.6 | 54.1 | 14.9 | - | 51.8 | 43.5 |
| skt | 40.3 | 25.5 | 34.7 | 15.2 | 38.5 | - | 30.8 |
| Avg | 31.1 | 28.3 | 29.9 | 12.7 | 33.4 | 35.6 | 28.5 |

$1.5 \times 10^9$ macs

| AnyDA | clp | inf | pnt | qdr | rel | skt | Avg |
|---|---|---|---|---|---|---|---|
| clp | - | 33.4 | 39.3 | 23.3 | 51.0 | 52.5 | 39.9 |
| inf | 17.3 | - | 17.4 | 2.8 | 20.1 | 17.3 | 15.0 |
| pnt | 36.2 | 32.2 | - | 7.4 | 49.9 | 42.7 | 33.7 |
| qdr | 12.4 | 4.4 | 4.9 | - | 7.2 | 14.3 | 8.6 |
| rel | 51.6 | 46.6 | 54.5 | 15.1 | - | 51.9 | 43.9 |
| skt | 40.5 | 25.2 | 35.3 | 15.2 | 38.9 | - | 31.0 |
| Avg | 31.6 | 28.4 | 30.3 | 12.8 | 33.4 | 35.7 | 28.7 |

$1.6 \times 10^9$ macs

| AnyDA | clp | inf | pnt | qdr | rel | skt | Avg |
|---|---|---|---|---|---|---|---|
| clp | - | 33.4 | 40.6 | 23.1 | 51.7 | 53.1 | 40.4 |
| inf | 18.2 | - | 17.1 | 3.8 | 21.3 | 17.6 | 15.6 |
| pnt | 37.8 | 33.2 | - | 8.2 | 51.2 | 42.9 | 34.7 |
| qdr | 12.2 | 4.8 | 4.8 | - | 7.6 | 14.2 | 8.7 |
| rel | 51.9 | 47.2 | 54.2 | 16.2 | - | 52.9 | 44.5 |
| skt | 41.8 | 26.1 | 38.9 | 17.5 | 39.6 | - | 32.8 |
| Avg | 32.4 | 28.9 | 31.1 | 13.8 | 34.3 | 36.1 | 29.4 |

$2.0 \times 10^9$ macs

| AnyDA | clp | inf | pnt | qdr | rel | skt | Avg |
|---|---|---|---|---|---|---|---|
| clp | - | 33.7 | 39.9 | 23.5 | 51.3 | 53.8 | 40.4 |
| inf | 18.3 | - | 17.5 | 3.9 | 21.7 | 17.6 | 15.8 |
| pnt | 37.6 | 33.6 | - | 8.9 | 51.8 | 44.0 | 35.2 |
| qdr | 12.0 | 4.1 | 4.8 | - | 7.4 | 13.6 | 8.4 |
| rel | 52.6 | 48.1 | 54.8 | 16.9 | - | 53.0 | 45.1 |
| skt | 41.9 | 27.1 | 39.9 | 17.9 | 39.5 | - | 33.3 |
| Avg | 32.5 | 29.3 | 31.4 | 14.2 | 34.3 | 36.4 | 29.7 |

