# OpenReview forum: "AnyDA: Anytime Domain Adaptation"
_ICLR.cc/2023/Conference — ICLR 2023 poster_

### Official Review · Reviewer_Z9Q2 · 2022-10-20

**Confidence:** 4
**Correctness:** 3
**Technical Novelty And Significance:** 3
**Empirical Novelty And Significance:** 3
**Recommendation:** 6

**Clarity, Quality, Novelty And Reproducibility:**

 quality of the paper is good.

 clarity of the paper is good.

and originality of the paper is good.


**Strength And Weaknesses:**

Firstly, I like the new concept proposed by this paper,  which considers domain alignment in addition to varying both network (width and depth) and input (resolution) scales to enable testing under a wide range of computation budgets. Such concept is quite pratical, since we need to apply different scales network on different terminals.

But also, although I like this concept, we feel it maybe a little bit like the mixture of several concepts. Just as mentioned in the paper, anytime network plus domain adaptation. What about first do the normal domain adaptation and then do the anytime inference?

And as for the method part, it is said that "the student minnet is trained to match its output with the average prediction of the teacher subnets, while the student subnets are trained to match their outputs with the prediction of the teacher supernet." why make this setting, I guess that it mainly for minimize the performance gap, but I don't think it would make much difference. And if it works, why don't make the same setting for the other subnet like the second small sub network?

And in this setting, I see that the authors set the teacher and the student network the same architecture. It is not flexible enough I think.

**Summary Of The Paper:**

The  paper propose a new concept, which is anytime domain adaptation.

Unlike the normal domain adaptation setting, the authors aim to make a generalised version of DA, which is tested under both at the level of input (resolution) and network (width, depth).

To achieve it,  the main idea of the paper is training two networks as teacher and student with switchable depth, width and input resolutions to enable testing under a wide range of computation budgets.

Extensive experiments are conducted on 4  benchmark datasets, and the most related work "slim DA" is also included for testing.

**Summary Of The Review:**

So my concerns are:

- What is the main contribution of this paper? What is the main difference between anytime network + da

- Why choose to match average performance? why not apply it to other subset networks?

- What about different architecture pairs.

---

> ### Author Response · Authors · 2022-11-17
> **Response to Reviewer Z9Q2 (Part 1)**
>
> e thank the reviewer for the thoughtful reviews and constructive suggestions.
>
> (a) **Key contributions and differences from directly combining existing domain adaptation and anytime prediction:** Our key contributions are: (1) we introduce a novel approach for anytime domain adaptation, that is executable with dynamic resource constraints to achieve accuracy-efficiency trade-offs under domain-shifts; (2) We propose a bootstrapped recursive distillation approach to train the student subnets with the knowledge from the teacher network that not only brings the target features close to the source but also transfers the learned knowledge to a smaller network for efficient inference.
>
> While existing works focus separately on anytime prediction and domain adaptation, we do not believe an ad-hoc combination of existing domain adaptation and anytime prediction can be helpful in this regard. Our novel design choices like (1) exposing the teacher network with data only from the target domain which encourages "target-to-source" correspondences for alleviating domain shift (refer Figure 2, top branch), (2) the use of a 'recursive' distillation technique, and (3) generating pseudo-labels from the student network in the proposed approach distinguish AnyDA from just being a combination of existing domain adaptation and anytime prediction techniques, rather an unique desing for learning domain invariant networks at multiple computation budgets. In addition, following reviewers suggestion, to further support our claims, we combine one existing state-of-the-art domain adaptation (DA) method, namely SymNet [1], with anytime prediction (AP) (US-Net [2] with slimmable width, depth and resolution as ours) naively in three settings:
>
>  (a.1) **Domain adaptation + Anytime prediction (inference only):** in this experiment, we first train a network using the SymNet domain adaptation method. Once trained, we used the adapted model directly for anytime inference to obtain its performance at various computation budgets on the target domain.
>
>  (a.2) **Domain adaptation + Anytime prediction (w/ Pseudo-labeling):** in this experiment, similar to point-(a.1) above, we obtain an adapted network using the SymNet domain adaptation technique on source and target data. After that, we further train the network in an anytime fashion using US-Net on the target data using self-supervision through pseudo-labels (PL). Note that since the target data is unlabeled, we need to use pseudo-labels to train on them using existing anytime networks.
>
>  (a.3) **Anytime prediction + Domain Adaptation:** in this experiment, we first train a network for anytime prediction on the labeled source data. Then, we train it using the SymNet domain adaptation approach.
>
> We perform the above experiments on Office-31 and Office-Home datasets and report the results in the tables below. We compare the performance with our proposed approach AnyDA using the same SymNet backbone network for fair comparison.
>
> | Office-31 |  DA + AP (inference only) | DA + AP (w/ PL) | AP + DA |AnyDA (ours) |
>  -------- | -------- | -------- | -------- | -------- |
> |1×|  86.4 |87.9|73.6|88.2 |
> |1/2×|35.7|43.6|46.3|87.9|
> |1/4×|31.8|38.5|43.2|87.8|
> |1/10×|30.0|37.7|40.0|87.2|
> |Average|46.0|51.9|50.8|87.8|
>
> | Office-Home |  DA + AP (inference only) | DA + AP (w/ PL) | AP + DA |AnyDA (ours) |
>  -------- | -------- | -------- | -------- | -------- |
> |1×| 62.8|62.7|56.5|68.7 |
> |1/2×|24.5|31.9|47.0|68.1|
> |1/4×|22.9|28.4|35.6|68.1|
> |1/10×|20.8|27.6|32.2|67.5|
> |Average|32.8|37.7|42.8|68.1|
>
> We have the following key observations from the above comparisons: (a.1) DA + AP (inference only): simply performing anytime inference on a domain adapted model performs poorly as compared to AnyDA even when using the full network with the highest budget (1.8% lower in Office-31, 5.9% lower in Office-Home) and fails miserably for the lower budget subnets (e.g. 57.2% lower in Office-31, 46.7% lower in Office-Home in the lowest budget configuration); (a.2) DA + AP (w/ PL): as can be seen, the performance improves as compared to point-(a.1) because of the additional anytime training on the target data. But, the the huge drop in performance from the highest budget network to the lower budget subnets is still significant (e.g. 49.5% and 39.9% lower than AnyDA in the lowest budget network); (a.3) While this variation performs better at lower budgets, the performance at the highest budget is lower than the other two variants, showing that anytime training without exploiting target data can give a poor initialization for anytime domain adaptation. Our approach outperforms AP+DA variant by 14.6% and 12.2% at the highest budget, while significantly outperforming it by more than 30% at the lower budgets.

---

> > ### Author Response · Authors · 2022-11-17
> > **Response to Reviewer Z9Q2 (Part 2)**
> >
> > To summarize, all the findings above corroborate the importance of the proposed components in AnyDA towards learning a robust network executable at different computation budget as well as alleviating the domain shift. We have included these additional results and anlysis in Appendix A of the revised draft.
> >
> > (b) **Matching average prediction for all the subnets:** An ensemble of teacher networks can generate more diverse, and accurate soft labels for distillation training of the student network. Thus, in our approach, the student minnet (lowest capacity) is trained to match its output with the average prediction of the teacher subnets such that the performance gap is minimized. On the other hand, the random student subnets (selected with sandwich rule) are trained to match their outputs with the prediction of the teacher supernet. We emperically test the effectiveness of this recursive distillation strategy in the ablation studies in Figure 4(b) and 4\(c\) on Office-31 dataset. In Figure 4(b), we consider encouraging all student subnets including the minnet to mimic the highest capacity teacher supernet and observe a consistent drop in performance across all budgets by around 1% compared to the proposed approach where the minnet learns from the average of the teacher subnets. We conjecture that the progressively lowering capacity and thus continually increasing gap of the student subnets with the teacher supernet can cause convergence hardship. Similarly, in Figure 4\(c\), we study the effect of mimicking the average of teacher soft logits for all student subnets and find that the performance drops by around 2% for all budgets showing that our recursive distillation strategy makes better use of the knowledge from multiple teacher subnets. We also observe a similar trend for Office-Home dataset as shown in Figure 10 of Appendix G in the revised draft.
> >
> > (c\) **Different teacher and student architecture:** Unlike conventional knowledge distillation between two networks of different sizes and architectures for model compression [3], our approach focuses on anytime domain adaptation where the goal is to flexibly adjust depth, width and input resolution of a **single deep network** during inference for instant adaptation in different scenarios under domain shifts. To achieve this, we adopt a teacher-student framework for bootstrapped recursive distillation where both teacher and student belong to the same network of a given architecture. In particular, *we build the teacher from past iterations of the student network as an exponential moving average (ema) of the student*. The bootstrapped teacher provides the targets to train the student for an enhanced representation. Thus, a different architecture pair is not a fit for the proposed AnyDA setup.
> >
> > **References:**
> >
> > [1] Rang Meng, Weijie Chen, Shicai Yang, Jie Song, Luojun Lin, Di Xie, Shiliang Pu, Xinchao Wang, Mingli Song, and Yueting Zhuang. Slimmable Domain Adaptation. CVPR, 2022.
> >
> > [2] Jiahui Yu and Thomas S Huang. Universally Slimmable Networks and Improved Training Techniques. ICCV, 2019.
> >
> > [3] Geoffrey Hinton, Oriol Vinyals, Jeff Dean. Distilling the Knowledge in a Neural Network. arXiv preprint arXiv:1503.02531, 2015.

---

### Official Review · Reviewer_Qxxc · 2022-10-20

**Confidence:** 2
**Correctness:** 3
**Technical Novelty And Significance:** 3
**Empirical Novelty And Significance:** 3
**Recommendation:** 6

**Clarity, Quality, Novelty And Reproducibility:**

The writing is mostly clear and the authors are suggested to public the codes for reproducibility.

**Strength And Weaknesses:**

1. Strength

(1) The task to tackle is a realistic setting that needs to adapt to different computational budgets in domain adaptation.

(2) The paper proposed a progressive student-teacher model to learn the sub-nets in different budgets. The proposed training loss is reasonable and demonstrates its effectiveness in DA benchmark datasets.

2. Weakness

(1) The basic idea of this work is simple and reasonable, however, combines the existing ideas of switchable sub-nets, BN, and the pseudo-label approach in domain adaptation.

(2) In Fig.3, the proposed approach is compared with the other methods, e.g., DDA, MSDNet. The curves show better performance when the budget is lower. Comparisons for larger computational budget should be also shown on these datasets.

(3) Since the training is based on distilling knowledge from larger sub-net to smaller sub-net. How does this training approach affect the higher capacity subnet. Will it may hurt the performance of the larger capacity subnet?

(4) The budget is pre-set, limiting its application if taking budget, might be out of range of pre-set budgets. How to handle this case? How does the competitors handle these out-of-range cases?

(5) How about the performance of extension to using larger backbones as teacher/student networks?

(6) How to understand the 'anytime' in the proposed setup.  'Anytime' may not be accurate to express the idea of DA adapted to different budgets.

**Summary Of The Paper:**

This paper proposed an domain adaptation method, adapted to different computational budgets when applying in the target domain. Specially, it considers the network depth, width and input resolutions in different budgets, and aims to learn a network that can be tailored to different network depth/width and input data sizes. The basic idea is based on the student-teacher framework, that distills the knowledge from larger capacity sub-networks to smaller capacity sub-networks. It also takes advantage of the switchable BNs and pseudo-label in the proposed method. The experiments show that the proposed method achieves higher performance than the DDA method, and marginally better than SlimDA.

**Summary Of The Review:**

The paper tackles a realistic problem in domain adaptation, and proposed an effective method for the anytime domain adaptation task. My major questions are on the more comparisons/evaluations, and clarifications on the extensible to out-of-range budget and larger backbone network.

---

> ### Author Response · Authors · 2022-11-17
> **Response to Reviewer Qxxc (Part 1)**
>
> We thank the reviewer for the thoughtful reviews and great suggestions. Below are our responses to the concerns and we have incorporated all the feedback in the revised version.
>
> (a) **Combination of existing ideas:** Our work forges a simple realistic connection between two literatures that have evolved mostly independently: anytime prediction and domain adaptation. The proposed bootstrapped recursive distillation adopts existing ideas like switchable BN and pseudo-labeling, in a non-trivial way as it trains the student subnets with the knowledge from the teacher network for not only bringing the target features close to the source but also transferring the learned knowledge to a smaller network for efficient inference. In particular, our novel design choices like (1) exposing the teacher network with data only from the target domain which encourages "target-to-source" correspondences for alleviating domain shift (refer Figure 2, top branch), (2) the use of a 'recursive' distillation technique, and (3) generating pseudo-labels from the student network in the proposed approach distinguish AnyDA from just being a combination of existing techniques, rather offer an unique design for learning domain invariant networks at multiple computation budgets.
>
> Moreover, in order to empirically support our claims, as suggested by reviewers, we combine one existing state-of-the-art domain adaptation (DA) method, namely SymNet [1], with anytime prediction (AP) (US-Net [2] with slimmable width, depth and resolution as ours) naively in three possible baseline settings on Office-31 and Office-Home, as discussed in Appendix C of the revised draft. Table 3 of Appendix C shows that a direct combination of these two existing methods fails miserably for the lower budget subnets even with additional pseudo-labeling added on top of it for harnessing categorical information from the target domain. This clearly corroborates the importance of the proposed components in AnyDA towards learning a robust network executable at different computation budget as well as alleviating the domain shift.
>
> (b) **Performance using larger backbone:** Figure 3 compares AnyDA using ResNet-50 with DDA(S4) which uses a multi-exit architecture built on top of MSDNet and supports only a 4-layer model limiting performance for higher budget values than provided. Following reviewer's suggestion, to compare at larger computational budget, we perform additional experiments using ResNet-101 on Office-31 dataset. Figure 5 in Appendix B shows the results. We train DDA using the largest available supported backbone of S7. As can be seen, AnyDA achieves the best average accuracy of 86.4% at a budget of 3.9x10^9 MACs. At the highest comparable budget with DDA of 1.6x10^9 MACs, AnyDA significantly outperforms by 4.2%. AnyDA's performance, unlike others, does not experience a drastic drop when the available budget is decreased. The improvement obtained at the lower budgets is even significantly more, _e.g.,_ at the corresponding lowest comparable budgets, AnyDA outperforms DDA by 16.2% at 0.3x10^9 MACs and ResNet by 5.9% at 0.9x10^9 MACs. This behavior clearly shows that our bootstrapped recursive distillation is not only able to train robust low-capacity models, but also ensures simultaneous domain alignment with larger backbones.
>
> (c\) **Effect of distillation on larger capacity subnets:** Thanks for the interesting question. As we are distilling from the larger subnet to smaller subnets, it is interesting to see if this has any adverse effect on the larger subnets. We investigate this by comparing accuracy at the largest budget of AnyDA with the ResNet baseline at the same budget for domain adaptation using two different backbones _e.g.,_ ResNet50 and ResNet101 on Office-31 dataset. The supernet with our bootstrapped recursive distillation gets an average performance boost of 2.3% (85.2% vs 82.9%: ref. Figure 3 in main paper) and 0.7% (86.4% vs 85.7%: ref. Figure 5 in Appendix B) compared to ResNet50 and ResNet101 at the same budget respectively. This shows that AnyDA does not adversely affect the bigger subnets, rather the synergistic cooperation between the subnets help the bigger networks to learn better domain invariant representation.

---

> > ### Author Response · Authors · 2022-11-17
> > **Response to Reviewer Qxxc (Part 2)**
> >
> > (d) **Out of range budgets:** In this work, we follow universally slimmable network (US-Net [2]) to set the range of budgets for all our experiments. US-Net focuses on training a single neural network executable at arbitrary budget within a range, unlike slimmable network [3] which is limited to  predefined budgets that it was trained on. Thus, specific budgets at which our network is executed during inference are not pre-set, instead only a range of budgets (minimum and maximum) for defining subnets during training is used in our current work. After training, our trained model (the student network) is executable at various budget configurations. The goal is to find the best configuration under a particular resource constraint. We achieve this by using a query table. For example, in ResNet50, we sample network width from {1, 0.9}, network depth from {1, 0.5} and sample input resolution from {224, 192, 160, 128}. We test all these width-depth-resolution configurations on a validation set and choose the best one under a given budget at inference. Since there is no re-training, the whole process is once for all. Our range of budget like others [2,4,5] is still bounded by the minimum and maximum MACs that the backbone network can support (e.g., maximum 2x10^9 MAC for ResNet50). For handling budgets that fall outside of this range, one can train a supernet with a very large backbone or by considering extreme low values of width, depth and resolution: we leave this as an interesting future work. We have added this discussions in Appendix F of the revised version.
> >
> > (e) **Understanding anytime in AnyDA:** We use the term "anytime" in our framework inspired from prior works on anytime prediction [6,7,8,9], where a _single network_ can give output by trading computation time for predictive accuracy by selecting from a set of candidate predictors. We focus on extending the anytime prediction in presence of domain shift and thus our approach is called Anytime Domain Adaptation (AnyDA in short). In particular, instead of conventional adaptation under a fixed computation budget, anytime in our work refers to directly running the model (training a model using both labeled source and unlabeled target data that) at arbitrary resource budget in the target domain while being invariant to distribution shifts across both domains. Our experiments show increase in performance as more and more computation is available with a very little drop at the lower budgets in comparison to the contemporary approaches.
> >
> >
> > **References:**
> >
> > [1] Yabin Zhang, Hui Tang, Kui Jia, and Mingkui Tan. Domain-Symmetric Networks for Adversarial Domain Adaptation. CVPR, 2019.
> >
> > [2] Jiahui Yu and Thomas S Huang. Universally Slimmable Networks and Improved Training Techniques. ICCV, 2019.
> >
> > [3] Jiahui Yu, Linjie Yang, Ning Xu, Jianchao Yang, and Thomas Huang. Slimmable neural networks. ICLR, 2019.
> >
> > [4] Taojiannan Yang, Sijie Zhu, Chen Chen, Shen Yan, Mi Zhang, and Andrew Willis. Mutualnet: Adaptive Convnet via Mutual Learning from Network Width and Resolution. ECCV, 2020.
> >
> > [5] Changlin Li, Guangrun Wang, Bing Wang, Xiaodan Liang, Zhihui Li, and Xiaojun Chang. Dynamic Slimmable Network. CVPR, 2021.
> >
> > [6] Gao Huang, Danlu Chen, Tianhong Li, Felix Wu, Laurens van der Maaten, Kilian Q. Weinberger. Multi-Scale Dense Networks for Resource Efficient Image Classification. ICLR, 2018.
> >
> > [7] Alex Grubb, Drew Bagnell. SpeedBoost: Anytime Prediction with Uniform Near-Optimality. AISTATS, 2012.
> >
> > [8] Han Cai, Chuang Gan, Tianzhe Wang, Zhekai Zhang and Song Han. Once-for-all: Train One Network and Specialize it for Efficient Deployment. ICLR, 2020.
> >
> > [9] Shlomo Zilberstein. Using Anytime Algorithms in Intelligent Systems. AI Magazine, 1996.

---

### Official Review · Reviewer_fqZv · 2022-10-24

**Confidence:** 4
**Correctness:** 3
**Technical Novelty And Significance:** 3
**Empirical Novelty And Significance:** 3
**Recommendation:** 8

**Clarity, Quality, Novelty And Reproducibility:**

This paper studies a novel domain adaption by combining with anytime prediction. The proposed method is novel and effective. The new problem and method indicate high quality and novelty of this paper. The proposed method is clearly explained and implementation details can ensure reproducibility.

**Strength And Weaknesses:**

Strengths:
1. The studied topic, i.e. anytime domain adaptation, is a novel domain adaptation setting with practical applications. The idea is interesting and the motivation is clearly explained.
2. The proposed method, i.e. teacher-student framework, with bootstrapped recursive distillation makes good sense and seems to be effective.
3. The experiments are convincing. The results are much better than the compared baselines. The ablation studies are promising.
4. The presentation is generally clear and fluent.

Weaknesses:
1. It needs more discussion on the limitations of directly combining existing domain adaptation and anytime prediction techniques.
2. The visualization of domain adaptation results is insufficient, such as t-SNE. I am interested to see how the visualization looks like under different computation budgets.
3. Some closely related reviews and surveys on domain adaptation are missing, such as "A review of domain adaptation without target labels", "A Review of Single-Source Deep Unsupervised Visual Domain Adaptation". The format of references is consisitent.
4. The analysis on the reasons that the proposed method can perform better than the baselines is not well summarized.


**Summary Of The Paper:**

This paper proposes to study a novel domain adaptation setting, i.e., anytime domain adaptation, by combining anytime prediction and domain adaptation. Specifically, two networks are trained as teacher and student with switchable depth, width, and input resolutions to enable testing under a wide range of computation budgets. A bootstrapped recursive distillation approach is designed to train the student subnets with the knowledge from the teacher network. In this way, the target features are brought close to the source and the learned knowledge can be better transferred to a smaller network for efficient inference. Experiments are conducted on four benchmark datasets to show the effectiveness of the proposed method.

**Summary Of The Review:**

Novel and interesting adaptation setting, effective method, convincing experiments and superior results but insufficient analysis, generally good presentation but inconsistent reference format.

---

> ### Author Response · Authors · 2022-11-17
> **Response to Reviewer fqZv (Part 1)**
>
> We thank Reviewer fqZv for confirming that our studied topic is a novel domain adaptation setting with practical applications, our proposed method is novel and effective, and our experiments are convincing with promising ablation studies.
>
> (a) **Directly combining existing domain adaptation and anytime prediction?:** As stated by the reviewer, learning domain invariant deployable deep neural networks is an important real-world problem with many practial applications. While existing works focus separately on anytime prediction and domain adaptation, we do not believe a naive combination of existing domain adaptation and anytime prediction techniques can be helpful in this regard. Our novel design choices like (1) exposing the teacher network with data only from the target domain which encourages "target-to-source" correspondences for alleviating domain shift (refer Figure 2, top branch), (2) the use of a 'recursive' distillation technique, and (3) generating pseudo-labels from the student network in the proposed approach distinguish AnyDA from just being a combination of existing domain adaptation and anytime prediction techniques, rather offers an unique design for learning domain invariant networks at multiple computation budgets. In addition, following reviewer's suggestion, to further support our claims, we combine one existing state-of-the-art domain adaptation (DA) method, namely SymNet [2], with anytime prediction (AP) (US-Net [1] with slimmable width, depth and resolution as ours) naively in three settings:
>
>  (a.1) **Domain adaptation + Anytime prediction (inference only):** in this experiment, we first train a network using the SymNet domain adaptation method. Once trained, we used the adapted model directly for anytime inference to obtain its performance at various computation budgets on the target domain.
>
> (a.2) **Domain adaptation + Anytime prediction (w/ Pseudo-labeling):** in this experiment, similar to point-(a.1) above, we obtain an adapted network using the SymNet domain adaptation technique on source and target data. After that, we further train the network in an anytime fashion using US-Net on the target data using self-supervision through pseudo-labels (PL). Note that since the target data is unlabeled, we need to use pseudo-labels to train on them using existing anytime networks.
>
>  (a.3) **Anytime prediction + Domain Adaptation:** in this experiment, we first train a network for anytime prediction on the labeled source data. Then, we train it using the SymNet domain adaptation approach.
>
> We perform the above experiments on Office-31 and Office-Home datasets and report the results in the tables below. We compare the performance with our proposed approach AnyDA using the same SymNet backbone network for fair comparison.
>
>
> | Office-31 |  DA + AP (inference only) | DA + AP (w/ PL) | AP + DA |AnyDA (ours) |
>  -------- | -------- | -------- | -------- | -------- |
> |1×|  86.4 |87.9|73.6|88.2 |
> |1/2×|35.7|43.6|46.3|87.9|
> |1/4×|31.8|38.5|43.2|87.8|
> |1/10×|30.0|37.7|40.0|87.2|
> |Average|46.0|51.9|50.8|87.8|
>
> | Office-Home |  DA + AP (inference only) | DA + AP (w/ PL) | AP + DA |AnyDA (ours) |
>  -------- | -------- | -------- | -------- | -------- |
> |1×| 62.8|62.7|56.5|68.7 |
> |1/2×|24.5|31.9|47.0|68.1|
> |1/4×|22.9|28.4|35.6|68.1|
> |1/10×|20.8|27.6|32.2|67.5|
> |Average|32.8|37.7|42.8|68.1|
>
> We have the following key observations from the above comparisons: (a.1) DA + AP (inference only): simply performing anytime inference on a domain adapted model performs poorly as compared to AnyDA even when using the full network with the highest budget (1.8% lower in Office-31, 5.9% lower in Office-Home) and fails miserably for the lower budget subnets (e.g. 57.2% lower in Office-31, 46.7% lower in Office-Home in the lowest budget configuration); (a.2) DA + AP (w/ PL): as can be seen, the performance improves as compared to point-(a.1) because of the additional anytime training on the target data. But, the the huge drop in performance from the highest budget network to the lower budget subnets is still significant (e.g. 49.5% and 39.9% lower than AnyDA in the lowest budget network); (a.3) While this variation performs better at lower budgets, the performance at the highest budget is lower than the other two variants, showing that anytime training without exploiting target data can give a poor initialization for anytime domain adaptation. Our approach outperforms AP+DA variant by 14.6% and 12.2% at the highest budget, while significantly outperforming it by more than 30% at the lower budgets for the Office-31 dataset. Similar observations can be made in the Office-Home dataset as well.
>
> To summarize, all the findings above corroborate the importance of the proposed components in AnyDA towards learning a robust network executable at different computation budget as well as alleviating the domain shift. We have included these additional results and anlysis in Appendix A of the revised draft.

---

> > ### Author Response · Authors · 2022-11-17
> > **Response to Reviewer fqZv (Part 2)**
> >
> > (b) **t-SNE visualization under different computation budgets:** Thanks for this suggestion! Figure 6 in Appendix C shows the t-SNE visualizations on four adaptation tasks (A->W, D->A, A->D and D->W) from the Office-31 dataset. The figure shows the clustering of the target features at various computation budget subnets from the *supernet* in the left to the *minnet* in the right for our approach. As can be seen the clustering is fairly consistent and discriminative across the subnets till the network with budget 0.6 x 10^9 MACs while slightly slackening for the very low budgets (after 5th column), showing the effectiveness of the bootstrapped recursive distillation in learning discriminative feature space at different budget configurations.
> >
> > (c\) **Missing references:** Thanks for pointing out the missing references, we have added them in the revised draft. We have also fixed the inconsistency in the format of the references.
> >
> > (d) **Summarizing performance improvements over baselines:** Our extensive experiments on 4 benchmark datasets show very minimal drop in performance across a wide range of computation budgets (refer Figure 3, Table 1 and Table 2). We have the following key observations while comparing our performance with different baselines: (i) AnyDA significantly outperforms MSDNet and ResNets mainly because of the effective distillation of knowledge for domain alignment as compared to naive domain adaptation on these architectures. (ii) Similarly when compared to REDA, DDA, and SlimDA, our approach overall trades the performance gracefully with decreasing budgets of the subnetworks. This is because AnyDA focuses on joint learning of robust low-budget networks and domain alignment through booststrapped recursive distillation while harnessing categorical information using pseudo-labels all at the same time. Moreover, AnyDA's variation over width, depth and resolution at the same time enables tighter as well as finer coupling of accuracy-efficiency than prior works that only focus on one or two out of the three dimensions. (iii) Finally, naive combinations of DA+AP as discussed above lead to drastically degraded performance in low budget networks owing to non-calibration between anytime prediction and domain alignment, unlike AnyDA which focuses on a synergistic combination of domain adaptation and anytime prediction to achieve best accuracy-efficiency trade-offs under domain-shifts.
> >
> > **References:**
> >
> > [1] Jiahui Yu and Thomas S Huang. Universally Slimmable Networks and Improved Training Techniques. ICCV, 2019.
> >
> > [2] Yabin Zhang, Hui Tang, Kui Jia, and Mingkui Tan. Domain-Symmetric Networks for Adversarial Domain Adaptation. CVPR, 2019.

---

### Official Review · Reviewer_yBmg · 2022-10-25

**Confidence:** 4
**Correctness:** 3
**Technical Novelty And Significance:** 3
**Empirical Novelty And Significance:** 2
**Recommendation:** 6

**Clarity, Quality, Novelty And Reproducibility:**

The paper is clearly written and provides a novel solution to infrequently studied problem area.
While there are questions of how sensitive the presented results are towards hyperparameters of the method, reproducing the algorithm itself should be straightforward from the presented work.

**Strength And Weaknesses:**

The paper tackles and underexplored problem area of anytime domain adaptation and presents an effective solution which substantively improves the performance of anytime networks over unsupervised target domains.
The paper is well written and easy to follow, with the exception of the sandwich rule, all components of the approach are intuitively motivated and empirically verified.

The main weaknesses of the paper is that the improvements over SlimDA appear to be marginal at best and its relationship with that work contemporary work is not well explained. Additional, the utility of anytime domain adaptation is not immediately evident as any system which employs it would necessarily need to expend substantial MACs in order to learn the adapted network for the target domain.
If the authors could directly address why the process of learning the adaptation network should not be considered in the need for efficient inference over a target dataset it would improve the quality of the work.

**Summary Of The Paper:**

Combining the interests to of two separate fields of inquiry the authors of this work tackle the problem of Anytime inference for Domain Adaptation through the use of a recursive knowledge distillation. Being able to perform inference over multiple computational budgets is a beneficial strategy for deployed machine learning approach and prior work has shown an undesirable sensitivity to domain shifts, this work proposes an useful extension which would allow models to perform anytime inference over shifted target domains.

**Summary Of The Review:**

The paper provides an interest approach to tackling the underexplored problem of anytime domain adaptation. The improvements over anytime baselines are consistent though not substantial and general utility of this problem area are not immediately apparent.
However, this work thorough and explores and interesting new direction, as we collectively work towards more reliable machine learning systems this work provides an interesting perspective.

---

> ### Author Response · Authors · 2022-11-17
> **Response to Reviewer yBmg (Part 1)**
>
> We thank Reviewer yBmg for acknowledging that our paper provides an interesting novel approach for tackling the underexplored problem of anytime domain adaptation. Below are our responses to the concerns and we have incorporated all the feedback/suggestions in the revised version.
>
> **(a) More details on sandwich rule:** Following universally slimmable networks (US-Net [1]), we use sandwich rule for training the network for better convergence behavior and overall performance. Specifically, in each iteration we train the model at minimum budget (*minnet*), highest budget (*supernet*) and 2 random intermediate budgets (*subnets*). This is inspired from the fact that performances at all budget are bounded by performance of the model at smallest budget (*minnet*) and largest budget (*supernet*) (see [1] for more details). Thus, optimizing performance lower bound and upper bound can implicitly optimize all subnetworks of different capacities. Additionally, this is computationally more efficient compared to training all the subnetworks in each iteration. We have added a brief description about this to make it clear in the revised version (see optimization in Section 3.3).
>
> **(b) Differences and improvements over SlimDA:** The key differences between our method and SlimDA lies at: **(i) Insight of reducing computation**. SlimDA adopts early-exit architectures with varying depth only, while we train a network with switchable depth, width and input resolution at the same time to enable tighter as well as finer coupling of efficiency-computation trade-off for inference over multiple budgets. **(ii) Generalization to any backbone**. SlimDA's exclusive dependence on a tri-classifier couples it strongly to a specific architecture, such as variations of symmetric adaptation network (SymNet [2]). In contrast, AnyDA fits equally well to any convnet architecture showing strong transferability across backbones (e.g., ResNet50 and ResNet101 including SymNets). Additionally, the Bayesian learning paradigm with Monte-Carlo approximation adapted by SlimDA depends on a good prior and this is shown by the use of a strong backbone like SymNet. Table 2 in the paper shows a comparision of AnyDA and SlimDA using SymNet backbone on 3 datasets. Our approach is better than SlimDA on all 3 datasets across various computation budgets while being competitive at lowest compared budget. Our average improvements over SlimDA are in the range of ~0.6% with an absolute improvement of 1.9% on Office-31 dataset at the full budget. We believe SymNet's good performances on the small-scale Office-31, Office-Home and ImageCLEF-DA may be a strong enabler for the saturating performances on them. Note that large-scale and challenging DomainNet dataset is a good testbed for the domain adaptation networks as demonstrated in the recent works [3,4,5]. On DomainNet, we are comprehensively beating DDA, the other contemporary work, showing the effectiveness of our approach for anytime domain adaptation on challenging adaptation scenarios. However, the unavailability of the Slim-DA code till now, did not allow us to compare the performance on this challenging dataset.
>
> **\(c\) Utility of anytime domain adaptation:** Thanks for this interesting question! The focus of our work is on learning a single network which is domain invariant and can be executed at different computational budgets (MACs) in multiple devices with different budget requirements. Training a single network with the ability to be executed at different budgets is more feasible and efficient than training separate networks of corresponding budget configurations. In many practical applications, once the model is trained, it is an extremely important to perform inference many times (without retraining) due to highly dynamic deployment environments (train once but inference many).
>
> We agree with the reviewer that computation cost of training a network for AnyDA is more than that of a conventional network due to joint training of all the subnets. However, we follow *sandwich rule* to reduce the training-time computation overhead by almost 4 times as compared to forward-passing through all the subnets. Focussing on both training-time as well as inference-time efficiency is an interesting research topic, which would be an exciting future work. We have added this discussion in Appendix D of the revised draft.

---

> > ### Author Response · Authors · 2022-11-17
> > **Response to Reviewer yBmg (Part 2)**
> >
> > **References:**
> >
> > [1] Jiahui Yu and Thomas S Huang. Universally Slimmable Networks and Improved Training Techniques. ICCV, 2019.
> >
> > [2] Yabin Zhang, Hui Tang, Kui Jia, and Mingkui Tan. Domain-Symmetric Networks for Adversarial Domain Adaptation. CVPR, 2019.
> >
> > [3] Jogendra Nath Kundu, Akshay R. Kulkarni, Suvaansh Bhambri, Deepesh Mehta, Shreyas Anand Kulkarni, Varun Jampani, and Venkatesh Babu Radhakrishnan. Balancing discriminability and transferability for source-free domain adaptation. ICML, 2022.
> >
> > [4] Dian Chen, Dequan Wang, Trevor Darrell and Sayna Ebrahimi. Contrastive Test-Time Adaptation. CVPR, 2022.
> >
> > [5] Christian Simon, Masoud Faraki, Yi-Hsuan Tsai, Xiang Yu, Samuel Schulter, Yumin Suh, Mehrtash Harandi and Manmohan Chandraker. On Generalizing Beyond Domains in Cross-Domain Continual Learning. CVPR, 2022.

---

### Author Response · Authors · 2022-11-17
**Summary of Author's Response and Paper Revision**

We would like to thank all the reviewers for their constructive comments! We are encouraged that reviewers find: (a) our work explores a new direction by presenting an interesting novel approach for tackling the underexplored problem of anytime domain adaptation, which has practical applications; (b) our teacher-student framework, with bootstrapped recursive distillation makes good sense, effective and intuitively motivated; \(c\) our experiments are convincing with promising ablation studies which shows consistent improvements over compared baselines.

We have addressed all the questions that the reviewers posed with additional experimental comparisons and clarifications. All of these additional experiments and suggestions have been added into the updated PDF (changes are highlighted in blue). Below, we summarize the main changes to the paper and request the reviewers to take a look at the new additions.

- Additional results on directly combining existing domain adaptation and anytime prediction, as suggested by R-fqZv and R-Z9Q2,

- t-SNE visualization under different computation budgets, as suggested by R-fqZv,

- Performance using larger backbone, as suggested by R-Qxxc,

- Clarification on sandwich rule, utility of AnyDA, as suggested by R-yBmg,

- Discussion on computational budgets and effect of distillation, as suggested by R-Qxxc,

- Discussion on matching average prediction and different teacher-student architectures, as suggested by R-Z9Q2.

---

> ### Comment · Reviewer_fqZv · 2022-12-06
> **for the authors response**
>
> I thank the authors for the response to my concerns. I agree with other reviewers on the strengths of this paper. My concerns are addressed in the response. I hope the authors would incorporate the updated results and analysis in the final version.

---

> > ### Author Response · Authors · 2022-12-06
> > **Thanks**
> >
> > Thanks for all the valuable feedback. We’re glad that our response addressed all your concerns. We have added all the additional experiments and suggestions into the updated PDF (changes are highlighted in blue).

---

### Decision · Program_Chairs · 2023-01-20

**Decision:**

Accept: poster

**Justification For Why Not Higher Score:**

I agree with the four reviewers that this paper is above the bar of ICLR. However, as pointed out the reviewers, the problem setting of this paper is within the scope of resource-efficient domain adaptation, which was studied by several pieces of previous work in 2020-2022. Thus the problem setting is not firstly initiated in this work as implied by the AnyDA title (kind of hyperbole). The technical part can be seen as a nontrivial integration of existing methods but with some crucial designs beneficial for empirical improvements. Note that, the current paper performs on par with a concurrent work SlimDA (CVPR 2022).

**Justification For Why Not Lower Score:**

All reviewers stand unanimously on the positive side with reasonable reviews. The paper is above the ICLR bar, free of technical flaws. AC has no grounds to override the decision.

**Metareview: Summary, Strengths And Weaknesses:**

This paper studies domain adaptation in resource-efficient application scenarios, with switchable depth, width and input resolutions on the fly to enable testing under a wide range of computation budgets. The approach, coined by Anytime Domain Adaptation (AnyDA), is a framework of bootstrapped recursive knowledge distillation that bridges the teacher network to the student network with switchable subnetworks. Empirical studies have been conducted on standard benchmarks, showing that AnyDA outperforms previous approaches such as REDA and DDA but performs on par with the concurrent approach SlimDA.

Four referees and the AC reviewed this paper with mixed initial ratings. After rebuttal and discussion, reviewers acknowledged that their concerns were addressed reasonably well, and the overall scores converged to be unanimous positive. AC considered the paper, the revision, the reviews, the rebuttal, and the discussion, and concurred with the reviewers.

However, AC has a personal suggestion regarding this particular subarea of domain adaptation: It is highly discouraged to claim new problem settings, such as Dynamic Domain Adaptation, Anytime Domain Adaptation, etc, given that these settings are essentially the same under the umbrella of "Resource-Efficient Domain Adaptation" (Jiang et al. 2020). It would be better to only initiate new settings unless they are essentially and substantially dissimilar. A better way is to claim a new approach towards the resource-efficient domain adaptation setting.

**Note From Pc:**

if the above contains the word "oral" or "spotlight" please see: "oral" presentation means -> notable-top-5% and "spotlight" means -> notable-top-25%. As stated in our emails, we are disassociating presentation type from AC recommendations